# DDPNOpt: Differential Dynamic Programming Neural Optimizer

**Guan-Horng Liu**[1][2]**, Tianrong Chen**[3]**, and Evangelos A. Theodorou**[1][2]
[1]Center for Machine Learning, [2]School of Aerospace Engineering,
[3]School of Electrical and Computer Engineering, Georgia Institute of Technology, USA
{ghliu,tianrong.chen,evangelos.theodorou}@gatech.edu

## Abstract

Interpretation of Deep Neural Networks (DNNs) training as an optimal control problem with nonlinear dynamical systems has received considerable attention recently, yet the algorithmic development remains relatively limited. In this work, we make an attempt along this line by reformulating the training procedure from the trajectory optimization perspective. We first show that most widely-used algorithms for training DNNs can be linked to the Differential Dynamic Programming (DDP), a celebrated second-order method rooted in the Approximate Dynamic Programming. In this vein, we propose a new class of optimizer, DDP Neural Optimizer (DDP-NOpt), for training feedforward and convolution networks. DDPNOpt features layer-wise feedback policies which improve convergence and reduce sensitivity to hyper-parameter over existing methods. It outperforms other optimal-control inspired training methods in both convergence and complexity, and is competitive against state-of-the-art first and second order methods. We also observe DDPNOpt has surprising benefit in preventing gradient vanishing. Our work opens up new avenues for principled algorithmic design built upon the optimal control theory.

## 1 Introduction

In this work, we consider the following optimal control problem (OCP) in the discrete-time setting:

$$\min_{\bar{\boldsymbol{u}}} J(\bar{\boldsymbol{u}}; \boldsymbol{x}_0) := \left[ \phi(\boldsymbol{x}_T) + \sum_{t=0}^{T-1} \ell_t(\boldsymbol{x}_t, \boldsymbol{u}_t) \right] \quad \text{s.t. } \boldsymbol{x}_{t+1} = f_t(\boldsymbol{x}_t, \boldsymbol{u}_t), \quad \text{(OCP)}$$

where $\boldsymbol{x}_t \in \mathbb{R}^n$ and $\boldsymbol{u}_t \in \mathbb{R}^m$ represent the state and control at each time step $t$. $f_t(\cdot, \cdot)$, $\ell_t(\cdot, \cdot)$ and $\phi(\cdot)$ respectively denote the nonlinear dynamics, intermediate cost and terminal cost functions. OCP aims to find a control trajectory, $\bar{\boldsymbol{u}} \triangleq \{\boldsymbol{u}_t\}_{t=0}^{T-1}$, such that the accumulated cost $J$ over the finite horizon $t \in \{0, 1, \cdots, T\}$ is minimized. Problems with the form of OCP appear in multidisciplinary areas since it describes a generic multi-stage decision making problem (Gamkrelidze, 2013), and have gained commensurate interest recently in deep learning (Weinan, 2017; Liu & Theodorou, 2019).

Central to the research along this line is the interpretation of DNNs as *discrete-time nonlinear dynamical systems*, where each layer is viewed as a distinct time step (Weinan, 2017). The dynamical system perspective provides a mathematically-sound explanation for existing DNN models (Lu et al., 2019). It also leads to new architectures inspired by numerical differential equations and physics (Lu et al., 2017; Chen et al., 2018; Greydanus et al., 2019). In this vein, one may interpret the training as the parameter identification (PI) of nonlinear dynamics. However, PI typically involves *(i)* searching time-independent parameters *(ii)* given trajectory measurements at each time step (Voss et al., 2004; Peifer & Timmer, 2007). Neither setup holds in piratical DNNs training, which instead optimizes time- (*i.e.* layer-) varying parameters given the target measurements only at the final stage.

An alternative perspective, which often leads to a richer analysis, is to recast network weights as *control variables*. Through this lens, OCP describes w.l.o.g. the training objective composed of layer-wise loss (*e.g.* weight decay) and terminal loss (*e.g.* cross-entropy). This perspective (see Table 1) has been explored recently to provide theoretical statements for convergence and generalization (Weinan et al., 2018; Seidman et al., 2020). On the algorithmic side, while OCP has motivated new

architectures (Benning et al., 2019) and methods for breaking sequential computation (Gunther et al., 2020; Zhang et al., 2019), OCP-inspired optimizers remain relatively limited, often restricted to either specific network class (*e.g.* discrete weight) (Li & Hao, 2018) or small-size dataset (Li et al., 2017).

The aforementioned works are primarily inspired by the Pontryagin Maximum Principle (PMP, Boltyanskii et al. (1960)), which characterizes the first-order optimality conditions to OCP. Another parallel methodology which receives little attention is the Approximate Dynamic Programming (ADP, Bertsekas et al. (1995)). Despite both originate from the optimal control theory, ADP differs from PMP in that at each time step a locally optimal *feedback policy* (as a function of state $\boldsymbol{x}_t$) is computed. These policies, as opposed to the vector update from PMP, are known to enhance the numerical stability of the optimization process when models admit chain structures (*e.g.* in DNNs) (Liao & Shoemaker, 1992; Tassa et al., 2012). Practical ADP algorithms such as the Differential Dynamic Programming (DDP, Jacobson & Mayne (1970)) appear extensively in modern autonomous systems for complex trajectory optimization (Tassa et al., 2014; Gu, 2017). However, whether they can be lifted to large-scale stochastic optimization, as in the DNN training, remains unclear.

In this work, we make a significant advance toward optimal-control-theoretic training algorithms inspired by ADP. We first show that most existing first- and second-order optimizers can be derived from DDP as special cases. Built upon this intriguing connection, we present a new class of optimizer which marries the best of both. The proposed method, DDP Neural Optimizer (DDPNOpt), features layer-wise feedback policies, which, as we will show through experiments, improve convergence and robustness. To enable efficient training, DDPNOpt adapts key components including *(i)* curvature adaption from existing methods, *(ii)* stabilization techniques used in trajectory optimization, and *(iii)* an efficient factorization to OCP. These lift the complexity by orders of magnitude compared with other OCP-inspired baselines, without sacrificing the performance. In summary, we present the following contributions.

Table 1: Terminology mapping

|  | Deep Learning | Optimal Control |
|---|---|---|
| $J$ | Total Loss | Trajectory Cost |
| $\boldsymbol{x}_t$ | Activation Vector | State Vector |
| $\boldsymbol{u}_t$ | Weight Parameter | Control Vector |
| $f$ | Layer Propagation | Dynamical System |
| $\phi$ | End-goal Loss | Terminal Cost |
| $\ell$ | Weight Decay | Intermediate Cost |

- We draw a novel perspective of DNN training from the trajectory optimization viewpoint, based on a theoretical connection between existing training methods and the DDP algorithm.

- We present a new class of optimizer, **DDPNOpt**, that performs a distinct backward pass inherited with Bellman optimality and generates layer-wise feedback policies to robustify the training against unstable hyperparameter (*e.g.* large learning rate) setups.

- We show that DDPNOpt achieves competitive performance against existing training methods on classification datasets and outperforms previous OCP-inspired methods in both training performance and runtime complexity. We also identify DDPNOpt can mitigate vanishing gradient.

## 2 PRELIMINARIES

We will start with the Bellman principle to OCP and leave discussions on PMP in Appendix A.1.

**Theorem 1 (Dynamic Programming (DP) (Bellman, 1954)).** *Define a value function $V_t : \mathbb{R}^n \mapsto \mathbb{R}$ at each time step that is computed backward in time using the Bellman equation*

$$V_t(\boldsymbol{x}_t) = \min_{\boldsymbol{u}_t(\boldsymbol{x}_t) \in \Gamma_{\boldsymbol{x}_t}} \underbrace{\ell_t(\boldsymbol{x}_t, \boldsymbol{u}_t) + V_{t+1}(f_t(\boldsymbol{x}_t, \boldsymbol{u}_t))}_{Q_t(\boldsymbol{x}_t, \boldsymbol{u}_t) \equiv Q_t}, \quad V_T(\boldsymbol{x}_T) = \phi(\boldsymbol{x}_T), \tag{1}$$

*where $\Gamma_{\boldsymbol{x}_t} : \mathbb{R}^n \mapsto \mathbb{R}^m$ denotes a set of mapping from state to control space. Then, we have $V_0(\boldsymbol{x}_0) = J^*(\boldsymbol{x}_0)$ be the optimal objective value to OCP. Further, let $\mu_t^*(\boldsymbol{x}_t) \in \Gamma_{\boldsymbol{x}_t}$ be the minimizer of Eq. 1 for each $t$, then the policy $\pi^* = \{\mu_t^*(\boldsymbol{x}_t)\}_{t=0}^{T-1}$ is globally optimal in the closed-loop system.*

---

**Notation:** We will always use $t$ as the time step of dynamics and denote a subsequence trajectory until time $s$ as $\bar{\boldsymbol{x}}_s \triangleq \{\boldsymbol{x}_t\}_{t=0}^s$, with $\bar{\boldsymbol{x}} \triangleq \{\boldsymbol{x}_t\}_{t=0}^T$ as the whole. For any real-valued time-dependent function $\mathcal{F}_t$, we denote its derivatives evaluated on a given state-control pair $(\boldsymbol{x}_t \in \mathbb{R}^n$ and $\boldsymbol{u}_t \in \mathbb{R}^m)$ as $\nabla_{\boldsymbol{x}_t}\mathcal{F}_t \in \mathbb{R}^n$, $\nabla_{\boldsymbol{x}_t}^2\mathcal{F}_t \in \mathbb{R}^{n \times n}$, $\nabla_{\boldsymbol{x}_t \boldsymbol{u}_t}\mathcal{F}_t \in \mathbb{R}^{n \times m}$, or simply $\mathcal{F}_{\boldsymbol{x}}^t$, $\mathcal{F}_{\boldsymbol{xx}}^t$, and $\mathcal{F}_{\boldsymbol{xu}}^t$ for brevity. The vector-tensor product, *i.e.* the contraction mapping on the dimension of the vector space, is denoted by $V_{\boldsymbol{x}} \cdot f_{\boldsymbol{xx}} \triangleq \sum_{i=1}^n [V_{\boldsymbol{x}}]_i [f_{\boldsymbol{xx}}]_i$, where $[V_{\boldsymbol{x}}]_i$ is the $i$-th element of the vector $V_{\boldsymbol{x}}$ and $[f_{\boldsymbol{xx}}]_i$ is the Hessian matrix corresponding to that element.

| **Algorithm 1** Differential Dynamic Programming | **Algorithm 2** Back-propagation (BP) with GD |
|---|---|
| 1: **Input:** $\bar{\boldsymbol{u}} \triangleq \{\boldsymbol{u}_t\}_{t=0}^{T-1}, \bar{\boldsymbol{x}} \triangleq \{\boldsymbol{x}_t\}_{t=0}^{T}$ | 1: **Input:** $\bar{\boldsymbol{u}} \triangleq \{\boldsymbol{u}_t\}_{t=0}^{T-1}, \bar{\boldsymbol{x}} \triangleq \{\boldsymbol{x}_t\}_{t=0}^{T}$, learning rate $\eta$ |
| 2: Set $V_{\boldsymbol{x}}^T = \nabla_{\boldsymbol{x}}\phi$ and $V_{\boldsymbol{xx}}^T = \nabla_{\boldsymbol{x}}^2\phi$ | 2: Set $\boldsymbol{p}_T \equiv \nabla_{\boldsymbol{x}_T} J_T = \nabla_{\boldsymbol{x}}\phi$ |
| 3: **for** $t = T-1$ **to** $0$ **do** | 3: **for** $t = T-1$ **to** $0$ **do** |
| 4:    Compute $\delta\boldsymbol{u}_t^*(\delta\boldsymbol{x}_t)$ using $V_{\boldsymbol{x}}^{t+1}, V_{\boldsymbol{xx}}^{t+1}$ (Eq. 3, 4) | 4:    $\delta\boldsymbol{u}_t^* = -\eta\nabla_{\boldsymbol{u}_t} J_t = -\eta(\ell_{\boldsymbol{u}}^t + f_{\boldsymbol{u}}^{t\ \mathsf{T}}\boldsymbol{p}_{t+1})$ |
| 5:    Compute $V_{\boldsymbol{x}}^t$ and $V_{\boldsymbol{xx}}^t$ using Eq. 5 | 5:    $\boldsymbol{p}_t \equiv \nabla_{\boldsymbol{x}_t} J_t = f_{\boldsymbol{x}}^{t\ \mathsf{T}}\boldsymbol{p}_{t+1}$ |
| 6: **end for** | 6: **end for** |
| 7: Set $\hat{\boldsymbol{x}}_0 = \boldsymbol{x}_0$ | 7: **for** $t = 0$ **to** $T-1$ **do** |
| 8: **for** $t = 0$ **to** $T-1$ **do** | 8:    $\boldsymbol{u}_t^* = \boldsymbol{u}_t + \delta\boldsymbol{u}_t^*$ |
| 9:    $\boldsymbol{u}_t^* = \boldsymbol{u}_t + \delta\boldsymbol{u}_t^*(\delta\boldsymbol{x}_t)$, where $\delta\boldsymbol{x}_t = \hat{\boldsymbol{x}}_t - \boldsymbol{x}_t$ | 9: **end for** |
| 10:    $\hat{\boldsymbol{x}}_{t+1} = f_t(\hat{\boldsymbol{x}}_t, \boldsymbol{u}_t^*)$ | 10: $\bar{\boldsymbol{u}} \leftarrow \{\boldsymbol{u}_t^*\}_{t=0}^{T-1}$ |
| 11: **end for** | |
| 12: $\bar{\boldsymbol{u}} \leftarrow \{\boldsymbol{u}_t^*\}_{t=0}^{T-1}$ | |

Hereafter we refer $Q_t(\boldsymbol{x}_t, \boldsymbol{u}_t)$ to the *Bellman objective*. The Bellman principle recasts minimization over a control sequence to a sequence of minimization over each control. The value function $V_t$ summarizes the optimal cost-to-go at each stage, provided all afterward stages also being minimized.

**Differential Dynamic Programming (DDP).** Despite providing the sufficient conditions to OCP, solving Eq. 1 for high-dimensional problems appears to be infeasible, well-known as the Bellman curse of dimensionality. To mitigate the computational burden of the minimization involved at each stage, one can approximate the Bellman objective in Eq. 1 with its second-order Taylor expansion. Such an approximation is central to DDP, a second-order trajectory optimization method that inherits a similar Bellman optimality structure while being computationally efficient.

Alg. 1 summarizes the DDP algorithm. Given a nominal trajectory $(\bar{\boldsymbol{x}}, \bar{\boldsymbol{u}})$ with its initial cost $J$, DDP iteratively optimizes the objective value, where each iteration consists a backward (lines 2-6) and forward pass (lines 7-11). During the backward pass, DDP performs second-order expansion on the Bellman objective $Q_t$ at each stage and computes the updates through the following minimization,

$$\delta\boldsymbol{u}_t^*(\delta\boldsymbol{x}_t) = \underset{\delta\boldsymbol{u}_t(\delta\boldsymbol{x}_t)\in\Gamma'_{\delta\boldsymbol{x}_t}}{\arg\min} \left\{ \frac{1}{2} \begin{bmatrix} \mathbf{1} \\ \delta\boldsymbol{x}_t \\ \delta\boldsymbol{u}_t \end{bmatrix}^{\mathsf{T}} \begin{bmatrix} \mathbf{0} & Q_{\boldsymbol{x}}^{t\ \mathsf{T}} & Q_{\boldsymbol{u}}^{t\ \mathsf{T}} \\ Q_{\boldsymbol{x}}^t & Q_{\boldsymbol{xx}}^t & Q_{\boldsymbol{xu}}^t \\ Q_{\boldsymbol{u}}^t & Q_{\boldsymbol{ux}}^t & Q_{\boldsymbol{uu}}^t \end{bmatrix} \begin{bmatrix} \mathbf{1} \\ \delta\boldsymbol{x}_t \\ \delta\boldsymbol{u}_t \end{bmatrix} \right\}, \tag{2}$$

$$\text{where} \quad \begin{aligned} Q_{\boldsymbol{x}}^t &= \ell_{\boldsymbol{x}}^t + f_{\boldsymbol{x}}^{t\ \mathsf{T}}V_{\boldsymbol{x}}^{t+1} \\ Q_{\boldsymbol{u}}^t &= \ell_{\boldsymbol{u}}^t + f_{\boldsymbol{u}}^{t\ \mathsf{T}}V_{\boldsymbol{x}}^{t+1} \end{aligned} , \quad \begin{aligned} Q_{\boldsymbol{xx}}^t &= \ell_{\boldsymbol{xx}}^t + f_{\boldsymbol{x}}^{t\ \mathsf{T}}V_{\boldsymbol{xx}}^{t+1}f_{\boldsymbol{x}}^t + V_{\boldsymbol{x}}^{t+1}\cdot f_{\boldsymbol{xx}}^t \\ Q_{\boldsymbol{uu}}^t &= \ell_{\boldsymbol{uu}}^t + f_{\boldsymbol{u}}^{t\ \mathsf{T}}V_{\boldsymbol{xx}}^{t+1}f_{\boldsymbol{u}}^t + V_{\boldsymbol{x}}^{t+1}\cdot f_{\boldsymbol{uu}}^t \\ Q_{\boldsymbol{ux}}^t &= \ell_{\boldsymbol{ux}}^t + f_{\boldsymbol{u}}^{t\ \mathsf{T}}V_{\boldsymbol{xx}}^{t+1}f_{\boldsymbol{x}}^t + V_{\boldsymbol{x}}^{t+1}\cdot f_{\boldsymbol{ux}}^t \end{aligned} . \tag{3}$$

We note that all derivatives in Eq. 3 are evaluated at the state-control pair $(\boldsymbol{x}_t, \boldsymbol{u}_t)$ at time $t$ among the nominal trajectory. The derivatives of $Q_t$ follow standard chain rule and the dot notation represents the product of a vector with a 3D tensor. $\Gamma'_{\delta\boldsymbol{x}_t} = \{\mathbf{b}_t + \mathbf{A}_t\delta\boldsymbol{x}_t : \mathbf{b}_t \in \mathbb{R}^m, \mathbf{A}_t \in \mathbb{R}^{m\times n}\}$ denotes the set of all *affine* mapping from $\delta\boldsymbol{x}_t$. The analytic solution to Eq. 2 admits a linear form given by

$$\delta\boldsymbol{u}_t^*(\delta\boldsymbol{x}_t) = \boldsymbol{k}_t + \boldsymbol{K}_t\delta\boldsymbol{x}_t \text{ , where } \boldsymbol{k}_t \triangleq -(Q_{\boldsymbol{uu}}^t)^{-1}Q_{\boldsymbol{u}}^t \text{ and } \boldsymbol{K}_t \triangleq -(Q_{\boldsymbol{uu}}^t)^{-1}Q_{\boldsymbol{ux}}^t \tag{4}$$

denote the open and feedback gains, respectively. $\delta\boldsymbol{x}_t$ is called the *state differential*, which will play an important role later in our analysis. Note that this policy is only optimal locally around the nominal trajectory where the second order approximation remains valid. Substituting Eq. 4 back to Eq. 2 gives us the backward update for $V_{\boldsymbol{x}}$ and $V_{\boldsymbol{xx}}$,

$$V_{\boldsymbol{x}}^t = Q_{\boldsymbol{x}}^t - Q_{\boldsymbol{ux}}^{t\ \mathsf{T}}(Q_{\boldsymbol{uu}}^t)^{-1}Q_{\boldsymbol{u}}^t , \quad \text{and} \quad V_{\boldsymbol{xx}}^t = Q_{\boldsymbol{xx}}^t - Q_{\boldsymbol{ux}}^{t\ \mathsf{T}}(Q_{\boldsymbol{uu}}^t)^{-1}Q_{\boldsymbol{ux}}^t . \tag{5}$$

In the forward pass, DDP applies the feedback policy sequentially from the initial time step while keeping track of the state differential between the new simulated trajectory and the nominal trajectory.

## 3   DIFFERENTIAL DYNAMIC PROGRAMMING NEURAL OPTIMIZER

### 3.1   TRAINING DNNS AS TRAJECTORY OPTIMIZATION

Recall that DNNs can be interpreted as dynamical systems where each layer is viewed as a distinct time step. Consider *e.g.* the propagation rule in feedforward layers,

$$\boldsymbol{x}_{t+1} = \sigma_t(\boldsymbol{h}_t), \quad \boldsymbol{h}_t = g_t(\boldsymbol{x}_t, \boldsymbol{u}_t) = \boldsymbol{W}_t\boldsymbol{x}_t + \boldsymbol{b}_t . \tag{6}$$

$x_t \in \mathbb{R}^{n_t}$ and $x_{t+1} \in \mathbb{R}^{n_{t+1}}$ represent the activation vector at layer $t$ and $t+1$, with $h_t \in \mathbb{R}^{n_{t+1}}$ being the pre-activation vector. $\sigma_t$ and $g_t$ respectively denote the nonlinear activation function and the affine transform parametrized by the vectorized weight $u_t \triangleq [\text{vec}(W_t), b_t]^\top$. Eq. 6 can be seen as a dynamical system (by setting $f_t \equiv \sigma_t \circ g_t$ in OCP) propagating the activation vector $x_t$ using $u_t$.

Next, notice that the gradient descent (GD) update, denoted $\delta\bar{u}^* \equiv -\eta \nabla_{\bar{u}} J$ with $\eta$ being the learning rate, can be break down into each layer, i.e. $\delta\bar{u}^* \triangleq \{\delta u_t^*\}_{t=0}^{T-1}$, and computed backward by

$$\delta u_t^* = \underset{\delta u_t \in \mathbb{R}^{m_t}}{\arg\min}\{J_t + \nabla_{u_t} J_t^\top \delta u_t + \tfrac{1}{2}\delta u_t^\top (\tfrac{1}{\eta} I_t)\delta u_t\}, \qquad (7)$$

$$\text{where } J_t(x_t, u_t) \triangleq \ell_t(u_t) + J_{t+1}(f_t(x_t, u_t), u_{t+1}), \quad J_T(x_T) \triangleq \phi(x_T) \qquad (8)$$

is the per-layer objective[1] at layer $t$. It can be readily verified that $p_t \equiv \nabla_{x_t} J_t$ gives the exact Back-propagation dynamics. Eq. 8 suggests that GD minimizes the quadratic expansion of $J_t$ with the Hessian $\nabla_{u_t}^2 J_t$ replaced by $\frac{1}{\eta} I_t$. Similarly, adaptive first-order methods, such as RMSprop and Adam, approximate the Hessian with the diagonal of the covariance matrix. Second-order methods, such as KFAC and EKFAC (Martens & Grosse, 2015; George et al., 2018), compute full matrices using Gauss-Newton (GN) approximation:

$$\nabla_u^2 J_t \approx \mathbb{E}[J_{u_t} J_{u_t}^\top] = \mathbb{E}[(x_t \otimes J_{h_t})(x_t \otimes J_{h_t})^\top] \approx \mathbb{E}[(x_t x_t^\top)] \otimes \mathbb{E}[(J_{h_t} J_{h_t}^\top)]. \qquad (9)$$

We now draw a novel connection between the training procedure of DNNs and DDP. Let us first summarize the Back-propagation (BP) with gradient descent in Alg. 2 and compare it with DDP (Alg. 1). At each training iteration, we treat the current weight as the control $\bar{u}$ that simulates the activation sequence $\bar{x}$. Starting from this nominal trajectory $(\bar{x}, \bar{u})$, both algorithms recursively define some layer-wise objectives ($J_t$ in Eq. 8 vs $V_t$ in Eq. 1), compute the weight/control update from the quadratic expansions (Eq. 7 vs Eq. 2), and then carry certain information ($\nabla_{x_t} J_t$ vs $(V_x^t, V_{xx}^t)$) backward to the preceding layer. The computation graph between the two approaches is summarized in Fig. 1. In the following proposition, we make this connection formally and provide conditions when the two algorithms become equivalent.

**Proposition 2.** *Assume $Q_{ux}^t = 0$ at all stages, then the backward dynamics of the value derivative can be described by the Back-propagation,*

$$\forall t, V_x^t = \nabla_{x_t} J, Q_u^t = \nabla_{u_t} J, Q_{uu}^t = \nabla_{u_t}^2 J. \qquad (10)$$

*In this case, the DDP policy is equivalent to the stage-wise Newton, in which the gradient is preconditioned by the block-wise inverse Hessian at each layer:*

$$k_t + K_t \delta x_t = -(\nabla_{u_t}^2 J)^{-1} \nabla_{u_t} J. \qquad (11)$$

*If further we have $Q_{uu}^t \approx \frac{1}{\eta} I$, then DDP degenerates to Back-propagation with gradient descent.*

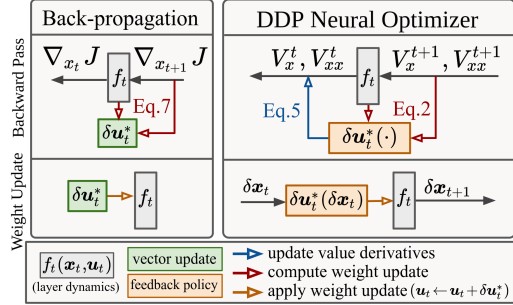

Figure 1: DDP backward propagates the value derivatives $(V_x, V_{xx})$ instead of $\nabla_{x_t} J$ and updates weight using layer-wise feedback policy, $\delta u_t^*(\delta x_t)$, with additional forward propagation.

Proof is left in Appendix A.2. Proposition 2 states that the backward pass in DDP collapses to BP when $Q_{ux}$ vanishes at all stages. In other words, existing training methods can be seen as special cases of DDP when the mixed derivatives (i.e. $\nabla_{x_t u_t}$) of the layer-wise objective are discarded.

## 3.2 EFFICIENT APPROXIMATION AND FACTORIZATION

Motivated by Proposition 2, we now present a new class of optimizer, DDP Neural Optimizer (DDPNOpt), on training feedforward and convolution networks. DDPNOpt follows the same procedure in vanilla DDP (Alg. 1) yet adapts several key traits arising from DNN training, which we highlight below.

**Evaluate derivatives of $Q_t$ with layer dynamics.** The primary computation in DDPNOpt comes from constructing the derivatives of $Q_t$ at each layer. When the dynamics is represented by the layer

---

[1] Hereafter we drop $x_t$ in all $\ell_t(\cdot)$ as the layer-wise loss typically involves weight regularization alone.

Table 2: Update rule at each layer $t$, $\boldsymbol{u}_t \leftarrow \boldsymbol{u}_t - \eta \boldsymbol{M}_t^{-1} \boldsymbol{d}_t$. (Expectation taken over batch data)

| Methods | Precondition matrix $\boldsymbol{M}_t$ | Update direction $\boldsymbol{d}_t$ |
|---|---|---|
| SGD | $\boldsymbol{I}_t$ | $\mathbb{E}[J_{\boldsymbol{u}_t}]$ |
| RMSprop | $\mathrm{diag}(\sqrt{\mathbb{E}[J_{\boldsymbol{u}_t} \odot J_{\boldsymbol{u}_t}]} + \epsilon)$ | $\mathbb{E}[J_{\boldsymbol{u}_t}]$ |
| KFAC & EKFAC | $\mathbb{E}[\boldsymbol{x}_t \boldsymbol{x}_t^\mathsf{T}] \otimes \mathbb{E}[J_{\boldsymbol{h}_t} J_{\boldsymbol{h}_t}^\mathsf{T}]$ | $\mathbb{E}[J_{\boldsymbol{u}_t}]$ |
| vanilla DDP | $\mathbb{E}[Q_{\boldsymbol{uu}}^t]$ | $\mathbb{E}[Q_{\boldsymbol{u}}^t + Q_{\boldsymbol{ux}}^t \delta \boldsymbol{x}_t]$ |
| **DDPNOpt** | $\boldsymbol{M}_t \in \left\{ \begin{array}{c} \boldsymbol{I}_t, \\ \mathrm{diag}(\sqrt{\mathbb{E}[Q_{\boldsymbol{u}}^t \odot Q_{\boldsymbol{u}}^t]} + \epsilon), \\ \mathbb{E}[\boldsymbol{x}_t \boldsymbol{x}_t^\mathsf{T}] \otimes \mathbb{E}[V_{\boldsymbol{h}}^t V_{\boldsymbol{h}}^{t\mathsf{T}}] \end{array} \right\}$ | $\mathbb{E}[Q_{\boldsymbol{u}}^t + Q_{\boldsymbol{ux}}^t \delta \boldsymbol{x}_t]$ |

propagation (recall Eq. 6 where we set $f_t \equiv \sigma_t \circ g_t$), we can rewrite Eq. 3 as:

$$Q_{\boldsymbol{x}}^t = g_{\boldsymbol{x}}^{t\,\mathsf{T}} V_{\boldsymbol{h}}^t, \quad Q_{\boldsymbol{u}}^t = \ell_{\boldsymbol{u}}^t + g_{\boldsymbol{u}}^{t\,\mathsf{T}} V_{\boldsymbol{h}}^t, \quad Q_{\boldsymbol{xx}}^t = g_{\boldsymbol{x}}^{t\,\mathsf{T}} V_{\boldsymbol{hh}}^t g_{\boldsymbol{x}}^t, \quad Q_{\boldsymbol{ux}}^t = g_{\boldsymbol{u}}^{t\,\mathsf{T}} V_{\boldsymbol{hh}}^t g_{\boldsymbol{x}}^t, \qquad (12)$$

where $V_{\boldsymbol{h}}^t \triangleq \sigma_{\boldsymbol{h}}^{t\,\mathsf{T}} V_{\boldsymbol{x}}^{t+1}$ and $V_{\boldsymbol{hh}}^t \triangleq \sigma_{\boldsymbol{h}}^{t\,\mathsf{T}} V_{\boldsymbol{xx}}^{t+1} \sigma_{\boldsymbol{h}}^t$ absorb the computation of the non-parametrized activation function $\sigma$. Note that Eq. 12 expands the dynamics only up to first order, *i.e.* we omitt the tensor products which involves second-order expansions on dynamics, as the stability obtained by keeping only the linearized dynamics is thoroughly discussed and widely adapted in practical DDP usages (Todorov & Li, 2005). The matrix-vector product with the Jacobian of the affine transform (*i.e.* $g_{\boldsymbol{u}}^t, g_{\boldsymbol{x}}^t$) can be evaluated efficiently for both feedforward (FF) and convolution (Conv) layers:

$$\boldsymbol{h}_t \overset{\mathrm{FF}}{=} \boldsymbol{W}_t \boldsymbol{x}_t + \boldsymbol{b}_t \Rightarrow g_{\boldsymbol{x}}^{t\,\mathsf{T}} V_{\boldsymbol{h}}^t = \boldsymbol{W}_t^\mathsf{T} V_{\boldsymbol{h}}^t, \quad g_{\boldsymbol{u}}^{t\,\mathsf{T}} V_{\boldsymbol{h}}^t = \boldsymbol{x}_t \otimes V_{\boldsymbol{h}}^t, \qquad (13)$$

$$\boldsymbol{h}_t \overset{\mathrm{Conv}}{=} \omega_t * \boldsymbol{x}_t \quad \Rightarrow g_{\boldsymbol{x}}^{t\,\mathsf{T}} V_{\boldsymbol{h}}^t = \omega_t^\mathsf{T} \mathbin{\hat{*}} V_{\boldsymbol{h}}^t, \quad g_{\boldsymbol{u}}^{t\,\mathsf{T}} V_{\boldsymbol{h}}^t = \boldsymbol{x}_t \mathbin{\hat{*}} V_{\boldsymbol{h}}^t, \qquad (14)$$

where $\otimes$, $\hat{*}$, and $*$ respectively denote the Kronecker product and (de-)convolution operator.

**Curvature approximation.** Next, since DNNs are highly over-parametrized models, $\boldsymbol{u}_t$ (*i.e.* the layer weight) will be in high-dimensional space. This makes $Q_{\boldsymbol{uu}}^t$ and $(Q_{\boldsymbol{uu}}^t)^{-1}$ computationally intractable to solve; thus requires approximation. Recall the interpretation we draw in Eq. 8 where existing optimizers differ in approximating the Hessian $\nabla_{\boldsymbol{u}_t}^2 J_t$. DDPNOpt adapts the same curvature approximation to $Q_{\boldsymbol{uu}}^t$. For instance, we can approximate $Q_{\boldsymbol{uu}}^t$ simply with an identity matrix $\boldsymbol{I}_t$, adaptive diagonal matrix $\mathrm{diag}(\sqrt{\mathbb{E}[Q_{\boldsymbol{u}}^t \odot Q_{\boldsymbol{u}}^t]})$, or the GN matrix:

$$Q_{\boldsymbol{uu}}^t \approx \mathbb{E}[Q_{\boldsymbol{u}}^t Q_{\boldsymbol{u}}^{t\,\mathsf{T}}] = \mathbb{E}[(\boldsymbol{x}_t \otimes V_{\boldsymbol{h}}^t)(\boldsymbol{x}_t \otimes V_{\boldsymbol{h}}^t)^\mathsf{T}] \approx \mathbb{E}[\boldsymbol{x}_t \boldsymbol{x}_t^\mathsf{T}] \otimes \mathbb{E}[V_{\boldsymbol{h}}^t V_{\boldsymbol{h}}^{t\,\mathsf{T}}]. \qquad (15)$$

Table 2 summarizes the difference in curvature approximation (*i.e.* the precondition $\boldsymbol{M}_t$) for different methods. Note that DDPNOpt constructs these approximations using $(V, Q)$ rather than $J$ since they consider different layer-wise objectives. As a direct implication from Proposition 2, DDPNOpt will degenerate to the optimizer it adapts for curvature approximation whenever all $Q_{\boldsymbol{ux}}^t$ vanish.

**Outer-product factorization.** When the memory efficiency becomes nonnegligible as the problem scales, we make GN approximation to $\nabla^2 \phi$, since the low-rank structure at the prediction layer has been observed for problems concerned in this work (Nar et al., 2019; Lezama et al., 2018). In the following proposition, we show that for a specific type of OCP, which happens to be the case of DNN training, such a low-rank structure preserves throughout the DDP backward pass.

**Proposition 3** (Outer-product factorization in DDPNOpt). *Consider the OCP where $\ell_t \equiv \ell_t(\boldsymbol{u}_t)$ is independent of $\boldsymbol{x}_t$. If the terminal-stage Hessian can be expressed by the outer product of vector $\boldsymbol{z}_{\boldsymbol{x}}^T$, $\nabla^2 \phi(\boldsymbol{x}_T) = \boldsymbol{z}_{\boldsymbol{x}}^T \otimes \boldsymbol{z}_{\boldsymbol{x}}^T$ (for instance, $\boldsymbol{z}_{\boldsymbol{x}}^T = \nabla \phi$ for GN), then we have the factorization for all $t$:*

$$Q_{\boldsymbol{ux}}^t = \boldsymbol{q}_{\boldsymbol{u}}^t \otimes \boldsymbol{q}_{\boldsymbol{x}}^t, \quad Q_{\boldsymbol{xx}}^t = \boldsymbol{q}_{\boldsymbol{x}}^t \otimes \boldsymbol{q}_{\boldsymbol{x}}^t, \quad V_{\boldsymbol{xx}}^t = \boldsymbol{z}_{\boldsymbol{x}}^t \otimes \boldsymbol{z}_{\boldsymbol{x}}^t. \qquad (16)$$

$\boldsymbol{q}_{\boldsymbol{u}}^t$, $\boldsymbol{q}_{\boldsymbol{x}}^t$, and $\boldsymbol{z}_{\boldsymbol{x}}^t$ are outer-product vectors which are also computed along the backward pass.

$$\boldsymbol{q}_{\boldsymbol{u}}^t = f_{\boldsymbol{u}}^{t\,\mathsf{T}} \boldsymbol{z}_{\boldsymbol{x}}^{t+1}, \quad \boldsymbol{q}_{\boldsymbol{x}}^t = f_{\boldsymbol{x}}^{t\,\mathsf{T}} \boldsymbol{z}_{\boldsymbol{x}}^{t+1}, \quad \boldsymbol{z}_{\boldsymbol{x}}^t = \sqrt{1 - \boldsymbol{q}_{\boldsymbol{u}}^{t\,\mathsf{T}} (Q_{\boldsymbol{uu}}^t)^{-1} \boldsymbol{q}_{\boldsymbol{u}}^t} \; \boldsymbol{q}_{\boldsymbol{x}}^t. \qquad (17)$$

The derivation is left in Appendix A.3. In other words, the outer-product factorization at the final layer can be backward propagated to all proceeding layers. Thus, large matrices, such as $Q_{\boldsymbol{ux}}^t$, $Q_{\boldsymbol{xx}}^t$, $V_{\boldsymbol{xx}}^t$, and even feedback policies $\boldsymbol{K}_t$, can be factorized accordingly, greatly reducing the complexity.

---

**Algorithm 3** Differential Dynamic Programming Neural Optimizer (DDPNOpt)

---

1: **Input:** dataset $\mathcal{D}$, learning rate $\eta$, training iteration $K$, batch size $B$, regularization $\epsilon_{V_{xx}}$
2: Initialize the network weights $\bar{u}^{(0)}$ (*i.e.* nominal control trajectory)
3: **for** $k = 1$ **to** $K$ **do**
4:      Sample batch initial state from dataset, $X_0 \equiv \{x_0^{(i)}\}_{i=1}^B \sim \mathcal{D}$
5:      Forward propagate to generate nominal batch trajectory $X_t$         $\triangleright$ Forward simulation
6:      Set $V_{x^{(i)}}^T = \nabla_{x^{(i)}} \Phi(x_T^{(i)})$ and $V_{xx^{(i)}}^T = \nabla_{x^{(i)}}^2 \Phi(x_T^{(i)})$
7:      **for** $t = T - 1$ **to** $0$ **do**                           $\triangleright$ Backward Bellman pass
8:          Compute $Q_u^t, Q_x^t, Q_{xx}^t, Q_{ux}^t$ with Eq. 12 (or Eq. 16-17 if factorization is used)
9:          Compute $\mathbb{E}[Q_{uu}^t]$ with one of the precondition matrices in Table 2
10:          Store the layer-wise feedback policy $\delta u_t^*(\delta X_t) = \frac{1}{B} \sum_{i=1}^B k_t^{(i)} + K_t^{(i)} \delta x_t^{(i)}$
11:          Compute $V_{x^{(i)}}^t$ and $V_{xx^{(i)}}^t$ with Eq. 5 (or Eq. 16-17 if factorization is used)
12:          $V_{xx^{(i)}}^t \leftarrow V_{xx^{(i)}}^t + \epsilon_{V_{xx}} I_t$ if regularization is used
13:      **end for**
14:      Set $\hat{x}_0^{(i)} = x_0^{(i)}$
15:      **for** $t = 0$ **to** $T - 1$ **do**                           $\triangleright$ Additional forward pass
16:          $u_t^* = u_t + \delta u_t^*(\delta X_t)$, where $\delta X_t = \{\hat{x}_t^{(i)} - x_t^{(i)}\}_{i=1}^B$
17:          $\hat{x}_{t+1}^{(i)} = f_t(\hat{x}_t^{(i)}, u_t^*)$
18:      **end for**
19:      $\bar{u}^{(k+1)} \leftarrow \{u_t^*\}_{t=0}^{T-1}$
20: **end for**

---

**Regularization on $V_{xx}$.** Finally, we apply Tikhonov regularization to the value Hessian $V_{xx}^t$ (line 12 in Alg. 3). This can be seen as placing a quadratic state-cost and has been shown to improve stability on optimizing complex humanoid behavior (Tassa et al., 2012). For the application of DNN where the dimension of the state (*i.e.* the vectorized activation) varies during forward/backward pass, the Tikhonov regularization prevents the value Hessian from low rank (throught $g_u^{t\top} V_{hh}^t g_x^t$); hence we also observe similar stabilization effect in practice.

## 4    THE ROLE OF FEEDBACK POLICIES

DDPNOpt differs from existing methods in the use of feedback $K_t$ and state differential $\delta x_t$. The presence of these terms result in a distinct backward pass inherited with the Bellman optimality. As shown in Table 2, the two frameworks differ in computing the update directions $d_t$, where the Bellman formulation applies the feedback policy through additional forward pass with $\delta x_t$. We have built the connection between these two $d_t$ in Proposition 2. In this section, we further characterize the role of the feedback policy $K_t$ and state differential $\delta x_t$ during optimization.

First we discuss the relation of DDPNOpt with other second-order methods and highlight the role feedback during training. To do so let us consider the example in Fig. 2a. Given an objective $L$ expanded at $(x_0, u_0)$, standard second-order methods compute the Hessian w.r.t. $u$ then apply the update $\delta u = -L_{uu}^{-1} L_u$ (shown as green arrows). DDPNOpt differs in that it also computes the *mixed* partial derivatives, *i.e.* $L_{ux}$. The resulting update law has the same intercept but with an additional feedback term linear in $\delta x$ (shown as red arrows). Thus, DDPNOpt searches for an update from the affine mapping $\Gamma'_{\delta x_t}$ (Eq. 2), rather than the vector space $\mathbb{R}^{m_t}$ (Eq. 7).

Next, to show how the state differential $\delta x_t$ arises during optimization, notice from Alg. 1 that $\hat{x}_t$ can be compactly expressed as $\hat{x}_t = F_t(x_0, \bar{u} + \delta \bar{u}^*(\delta \bar{x}))^2$. Therefore, $\delta x_t = \hat{x}_t - x_t$ captures the state difference when new updates $\delta \bar{u}^*(\delta \bar{x})$ are applied until layer $t - 1$. Now, consider the 2D example in Fig 2b. Back-propagation proposes the update directions (shown as blue arrows) from the first-order derivatives expanded along the nominal trajectory $(\bar{x}, \bar{u})$. However, as the weight at each layer is correlated, parameter updates from previous layers $\delta \bar{u}_s^*$ affect proceeding states $\{x_t : t > s\}$, thus the trustworthiness of their descending directions. As shown in Fig 2c, cascading these (green) updates may cause an *over-shoot* w.r.t. the designed target. From the trajectory optimization perspective, a much stabler direction will be instead $\nabla_{u_t} J_t(\hat{x}_t, u_t)$ (shown as orange), where the derivative is

---

[2] $F_t \triangleq f_t \circ \cdots \circ f_0$ denotes the compositional dynamics propagating $x_0$ with the control sequence $\{u_s\}_{s=0}^t$.

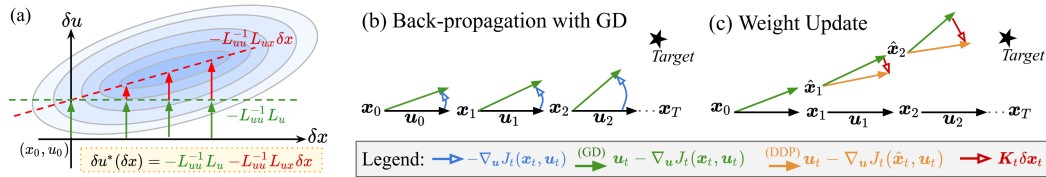

Figure 2: (a) A toy illustration of the standard update (green) and the DDP feedback (red). The DDP policy in this case is a line lying at the valley of objective $L$. (bc) Trajectory optimization viewpoint of DNN training. Green and orange arrows represent the proposed updates from GD and DDP.

evaluated at the new cascading state $\hat{\boldsymbol{x}}_t$, which accounts for previous updates, rather than the original state $\boldsymbol{x}_t$. This is exactly what DDPNOpt proposes, as we can derive the relation (see Appendix A.5),

$$\boldsymbol{K}_t \delta \boldsymbol{x}_t \approx \underset{\delta \boldsymbol{u}_t(\delta \boldsymbol{x}_t) \in \Gamma'_{\delta \boldsymbol{x}_t}}{\arg \min} \|\nabla_{\boldsymbol{u}_t} J(\hat{\boldsymbol{x}}_t, \boldsymbol{u}_t + \delta \boldsymbol{u}_t(\delta \boldsymbol{x}_t)) - \nabla_{\boldsymbol{u}_t} J(\boldsymbol{x}_t, \boldsymbol{u}_t)\| \ . \tag{18}$$

Thus, the feedback direction compensates the over-shoot by steering the GD update toward $\nabla_{\boldsymbol{u}_t} J_t(\hat{\boldsymbol{x}}_t, \boldsymbol{u}_t)$ after observing $\delta \boldsymbol{x}_t$. The difference between $\nabla_{\boldsymbol{u}_t} J(\hat{\boldsymbol{x}}_t, \boldsymbol{u}_t)$ and $\nabla_{\boldsymbol{u}_t} J(\boldsymbol{x}_t, \boldsymbol{u}_t)$ cannot be neglected especially during early training when the loss landscape contains nontrivial curvature everywhere (Alain et al., 2019). In short, the use of feedback $\boldsymbol{K}_t$ and state differential $\delta \boldsymbol{x}_t$ arises from the fact that *deep nets exhibit chain structures*. DDPNOpt feedback policies thus have a stabilization effect on robustifying the training dynamics against *e.g.* improper hyper-parameters which may cause unstable training. This perspective (*i.e.* optimizing chained parameters) is explored rigorously in trajectory optimization, where DDP is shown to be numerically stabler than direct optimization such as Newton method (Liao & Shoemaker, 1992).

**Remarks on other optimizers.** Our discussions so far rigorously explore the connection between DDP and stage/layer-wise Newton, thus include many popular second-order training methods. General Newton method coincides with DDP only for linear dynamics (Murray & Yakowitz, 1984), despite both share the same convergence rate when the dynamics is fully expanded to second order. We note that computing layer-wise value Hessians with only first-order expansion on the dynamics (Eq. 12) resembles the computation in Gauss-Newton method (Botev et al., 2017). For other control-theoretic methods, *e.g.* PID optimizers (An et al., 2018), they mostly consider the dynamics over training iterations. DDPNOpt instead focuses on the dynamics inherited in the DNN architecture.

## 5 EXPERIMENTS

### 5.1 PERFORMANCE ON CLASSIFICATION DATASET

**Networks & Baselines Setup.** We first validate the performance of training fully-connected (FCN) and convolution networks (CNN) using DDPNOpt on classification datasets. FCN consists of 5 fully-connected layers with the hidden dimension ranging from 10 to 32, depending on the size of the dataset. CNN consists of 4 convolution layers (with 3×3 kernel, 32 channels), followed by 2 fully-connected layers. We use ReLU activation on all datasets except Tanh for WINE and DIGITS to better distinguish the differences between optimizers. The batch size is set to 8-32 for datasets trained with FCN, and 128 for datasets trained with CNN. As DDPNOpt combines strengths from both standard training methods and OCP framework, we select baselines from both sides. This includes first-order methods, *i.e.* SGD (with tuned momentum), RMSprop, Adam, and second-order method EKFAC (George et al., 2018), which is a recent extension of the popular KFAC (Martens & Grosse, 2015). For OCP-inspired methods, we compare DDPNOpt with vanilla DDP and E-MSA (Li et al., 2017), which is also a second-order method yet built upon the PMP framework. Regarding the curvature approximation used in DDPNOpt ($\boldsymbol{M}_t$ in Table 2), we found that using adaptive diagonal and GN matrices respectively for FCNs and CNNs give the best performance in practice. We leave the complete experiment setup and additional results in Appendix A.6.

**Training Results.** Table 3 presents the results over 10 random trials. It is clear that DDPNOpt outperforms two OCP baselines on *all datasets and network types*. In practice, both baselines suffer from unstable training and require careful tuning on the hyper-parameters. In fact, we are not able to obtain results for vanilla DDP with any reasonable amount of computational resources when the problem size goes beyond FC networks. This is in contrast to DDPNOpt which adapts amortized

Table 3: Performance comparison on accuracy (%). All values averaged over 10 seeds.

| | DataSet | Standard baselines | | | | OCP-inspired baselines | | DDPNOpt (ours) |
| | | SGD-m | RMSProp | Adam | EKFAC | E-MSA | vanilla DDP | |
|---|---|---|---|---|---|---|---|---|
| Feed-forward | WINE | 94.35 | 98.10 | 98.13 | 94.60 | 93.56 | 98.00 | **98.18** |
| | DIGITS | **95.36** | 94.33 | 94.98 | 95.24 | 94.87 | 91.68 | 95.13 |
| | MNIST | 92.65 | 91.89 | 92.54 | 92.73 | 90.24 | 90.42 | **93.30** |
| | F-MNIST | 82.49 | 83.87 | 84.36 | 84.12 | 82.04 | 81.98 | **84.98** |
| CNN | MNIST | 97.94 | 98.05 | 98.04 | 98.02 | 96.48 | N/A | **98.09** |
| | SVHN | 89.00 | 88.41 | 87.76 | 90.63 | 79.45 | | **90.70** |
| | CIFAR-10 | 71.26 | 70.52 | 70.04 | 71.85 | 61.42 | | **71.92** |

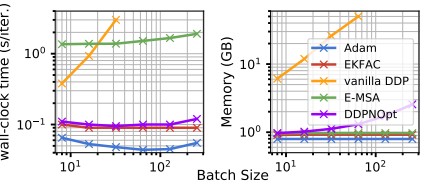

Figure 3: Runtime comparison on MNIST.

Table 4: Computational complexity in backward pass. ($B$: batch size, $X$: hidden state dim., $L$: # of layers)

| Method | Adam | Vanilla DDP | **DDPNOpt** |
|---|---|---|---|
| Memory | $\mathcal{O}(X^2 L)$ | $\mathcal{O}(BX^3 L)$ | $\mathcal{O}(X^2 L + BX)$ |
| Speed | $\mathcal{O}(BX^2 L)$ | $\mathcal{O}(B^3 X^3 L)$ | $\mathcal{O}(BX^2 L)$ |

curvature estimation from widely-used methods; thus exhibits much stabler training dynamics with superior convergence. In Table 4, we provide the analytic runtime and memory complexity among different methods. While vanilla DDP grows cubic w.r.t. $BX$, DDPNOpt reduces the computation by orders of magnitude with efficient approximation presented in Sec. 3. As a result, when measuring the actual computational performance with GPU parallelism, DDPNOpt runs nearly as fast as standard methods and outperforms E-MSA by a large margin. The additional memory complexity, when comparing DDP-inspired methods with Back-propagation methods, comes from the layer-wise feedback policies. However, DDPNOpt is much memory-efficient compared with vanilla DDP by exploiting the factorization in Proposition 3.

**Ablation Analysis.** On the other hand, the performance gain between DDPNOpt and standard methods appear comparatively small. We conjecture this is due to the inevitable use of similar curvature adaptation, as the local geometry of the landscape directly affects the convergence behavior. To identify scenarios where DDPNOpt best shows its effectiveness, we conduct an ablation analysis on the feedback mechanism. This is done by recalling Proposition 2: when $Q_{ux}^t$ vanishes, DDPNOpt degenerates to the method associated with each precondition matrix. For instance, DDPNOpt with identity (*resp.* adaptive diagonal and GN) precondition $M_t$ will generate the same updates as SGD (*resp.* RMSprop and EKFAC) when all $Q_{ux}^t$ are zeroed out. In other words, these DDPNOpt variants can be viewed as the *DDP-extension* to existing baselines.

In Fig. 4a we report the performance difference between each baseline and its associated DDPNOpt variant. Each grid corresponds to a distinct training configuration that is averaged over 10 random trails, and we keep all hyper-parameters (*e.g.* learning rate and weight decay) the same between baselines and their DDPNOpt variants. Thus, the performance gap only comes from the feedback policies, or equivalently the update directions in Table 2. Blue (*resp.* red) indicates an improvement (*resp.* degradation) when the feedback policies are presented. Clearly, the improvement over baselines remains consistent across most hyper-parameters setups, and the performance gap tends to become obvious as the learning rate increases. This aligns with the previous study on numerical stability (Liao & Shoemaker, 1992), which suggests the feedback can stabilize the optimization when *e.g.* larger control updates are taken. Since larger control corresponds to a further step size in the application of DNN training, one should expect DDPNOpt to show its robustness as the learning rate increases. As shown in Fig. 4b, such a stabilization can also lead to smaller variance and faster convergence. This sheds light on the benefit gained by bridging two seemly disconnected methodologies between DNN training and trajectory optimization.

## 5.2 DISCUSSION ON FEEDBACK POLICIES

**Visualization of Feedback Policies.** To understand the effect of feedback policies more perceptually, in Fig. 5 we visualize the feedback policy when training CNNs. This is done by first conducting

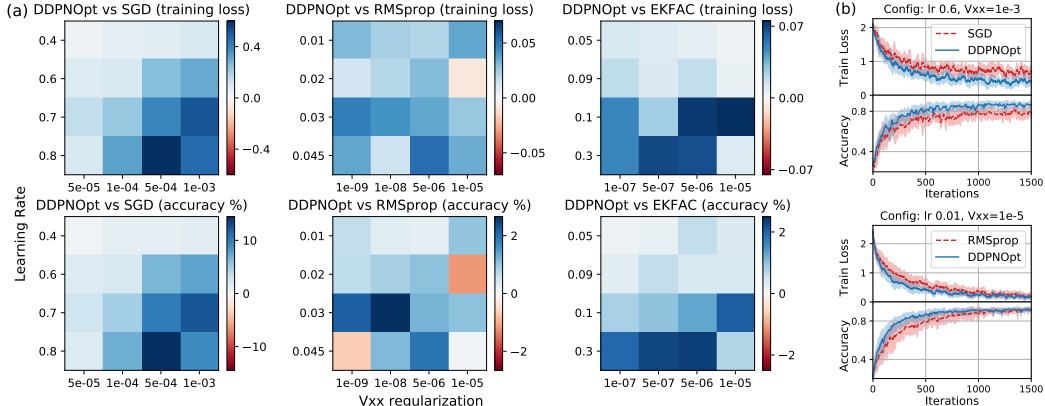

Figure 4: (a) Performance difference between DDPNOpt and baselines on DIGITS across hyper-parameter grid. Blue (*resp.* red) indicates an improvement (*resp.* degradation) over baselines. We observe similar behaviors on other datasets. (b) Examples of the actual training dynamics.

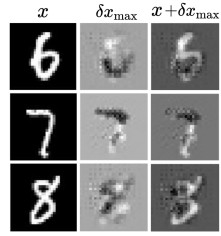

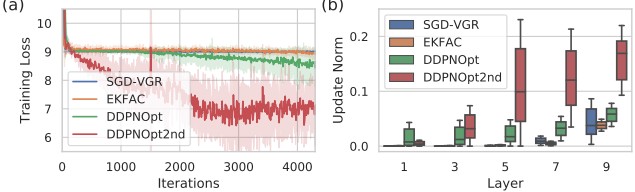

Figure 5: Visualization of the feedback policies on MNIST.

Figure 6: Training a 9-layer sigmoid-activated FCN on DIGITS using MMC loss. DDPNOpt2nd denotes when the layer dynamics is fully expanded to the second order.

singular-value decomposition on the feedback matrices $K_t$, then projecting the leading right-singular vector back to image space (see Alg. 4 and Fig. 7 in Appendix for the pseudo-code). These feature maps, denoted $\delta x_{\max}$ in Fig. 5, correspond to the dominating differential image that the policy shall respond with during weight update. Fig. 5 shows that the feedback policies indeed capture non-trivial visual features related to the pixel-wise difference between spatially similar classes, *e.g.* $(8, 3)$ or $(7, 1)$. These differential maps differ from adversarial perturbation (Goodfellow et al., 2014) as the former directly links the parameter update to the change in activation; thus being more interpretable.

**Vanishing Gradient.** Lastly, we present an interesting finding on how the feedback policies help mitigate vanishing gradient (VG), a notorious effect when DNNs become impossible to train as gradients vanish along Back-propagation. Fig. 6a reports results on training a sigmoid-activated DNN on DIGITS. We select SGD-VGR, which imposes a specific regularization to mitigate VG (Pascanu et al., 2013), and EKFAC as our baselines. While both baselines suffer to make any progress, DDPNOpt continues to generate non-trivial updates as the state-dependent feedback, *i.e.* $K_t \delta x_t$, remains active. The effect becomes significant when dynamics is fully expanded to the second order. As shown in Fig. 6b, the update norm from DDPNOpt is typically 5-10 times larger. We note that in this experiment, we replace the cross-entropy (CE) with Max-Mahalanobis center (MMC) loss, a new classification objective that improves robustness on standard vision datasets (Pang et al., 2019). MMC casts classification to distributional regression, providing denser Hessian and making problems similar to original trajectory optimization. None of the algorithms escape from VG using CE. We highlight that while VG is typically mitigated on the *architecture* basis, by having either unbounded activation function or residual blocks, DDPNOpt provides an alternative from the *algorithmic* perspective.

## 6 CONCLUSION

In this work, we introduce DDPNOpt, a new class of optimizer arising from a novel perspective by bridging DNN training to optimal control and trajectory optimization. DDPNOpt features layer-wise feedback policies which improve convergence and robustness to hyper-parameters over existing optimizers. It outperforms other OCP-inspired methods in both training performance and scalability. This work provides a new algorithmic insight and bridges between deep learning and optimal control.

## ACKNOWLEDGMENTS

The authors would like to thank Chen-Hsuan Lin, Yunpeng Pan, Yen-Cheng Liu, and Chia-Wen Kuo for many helpful discussions on the paper. This research was supported by NSF Award Number 1932288.

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

## A   APPENDIX

### A.1   CONNECTION BETWEEN PONTRYAGIN MAXIMUM PRINCIPLE AND DNNS TRAINING

Development of the optimality conditions to OCP can be dated back to 1960s, characterized by both the Pontryagin's Maximum Principle (PMP) and the Dynamic Programming (DP). Here we review Theorem of PMP and its connection to training DNNs.

**Theorem 4** (Discrete-time PMP (Pontryagin et al., 1962)). *Let $\bar{u}^*$ be the optimal control trajectory for OCP and $\bar{x}^*$ be the corresponding state trajectory. Then, there exists a co-state trajectory $\bar{p}^* \triangleq \{p_t^*\}_{t=1}^T$, such that*

$$x_{t+1}^* = \nabla_p H_t\left(x_t^*, p_{t+1}^*, u_t^*\right), \; x_0^* = x_0\,, \tag{19a}$$

$$p_t^* = \nabla_x H_t\left(x_t^*, p_{t+1}^*, u_t^*\right), \; p_T^* = \nabla_x \phi\left(x_T^*\right)\,, \tag{19b}$$

$$u_t^* = \operatorname*{arg\,min}_{v \in \mathbb{R}^m} H_t\left(x_t^*, p_{t+1}^*, v\right)\,. \tag{19c}$$

*where $H_t : \mathbb{R}^n \times \mathbb{R}^n \times \mathbb{R}^m \mapsto \mathbb{R}$ is the discrete-time Hamiltonian given by*

$$H_t\left(x_t, p_{t+1}, u_t\right) \triangleq \ell_t(x_t, u_t) + p_{t+1}^\mathsf{T} f_t(x_t, u_t)\,, \tag{20}$$

*and Eq. 19b is called the adjoint equation.*

The discrete-time PMP theorem can be derived using KKT conditions, in which the co-state $p_t$ is equivalent to the Lagrange multiplier. Note that the solution to Eq. 19c admits an open-loop process in the sense that it does not depend on state variables. This is in contrast to the Dynamic Programming principle, in which a feedback policy is considered.

It is natural to ask whether the necessary condition in the PMP theorem relates to first-order optimization methods in DNN training. This is indeed the case as pointed out in Li et al. (2017):

**Lemma 5** (Li et al. (2017)). *Back-propagation satisfies Eq. 19b and gradient descent iteratively solves Eq. 19c.*

Lemma 5 follows by first expanding the derivative of Hamiltonian w.r.t. $x_t$,

$$\nabla_{x_t} H_t(x_t, p_{t+1}, u_t) = \nabla_{x_t} \ell_t(x_t, u_t) + \nabla_{x_t} f_t(x_t, u_t)^\mathsf{T} p_{t+1} \; = \nabla_{x_t} J(\bar{u}; x_0)\,. \tag{21}$$

Thus, Eq. 19b is simply the chain rule used in the Back-propagation. When $H_t$ is differentiable w.r.t. $u_t$, one can attempt to solve Eq. 19c by iteratively taking the gradient descent. This will lead to

$$u_t^{(k+1)} = u_t^{(k)} - \eta \nabla_{u_t} H_t(x_t, p_{t+1}, u_t) = u_t^{(k)} - \eta \nabla_{u_t} J(\bar{u}; x_0)\,, \tag{22}$$

where $k$ and $\eta$ denote the update iteration and step size. Thus, existing optimization methods can be interpreted as iterative processes to match the PMP optimality conditions.

Inspired from Lemma 5, Li et al. (2017) proposed a PMP-inspired method, named Extended Method of Successive Approximations (E-MSA), which solves the following augmented Hamiltonian

$$\tilde{H}_t\left(x_t, p_{t+1}, u_t, x_{t+1}, p_t\right) \triangleq H_t\left(x_t, p_{t+1}, u_t\right)$$
$$+ \frac{1}{2}\rho \left\| x_{t+1} - f_t(x_t, u_t) \right\| + \frac{1}{2}\rho \left\| p_t - \nabla_{x_t} H_t \right\|\,. \tag{23}$$

$\tilde{H}_t$ is the original Hamiltonian augmented with the feasibility constraints on both forward states and backward co-states. E-MSA solves the minimization

$$\boldsymbol{u}_t^* = \underset{\boldsymbol{u}_t \in \mathbf{R}^{m_t}}{\arg\min} \tilde{H}_t \left( \boldsymbol{x}_t, \boldsymbol{p}_{t+1}, \boldsymbol{u}_t, \boldsymbol{x}_{t+1}, \boldsymbol{p}_t \right) \tag{24}$$

with L-BFGS per layer and per training iteration. As a result, we consider E-MSA also as second-order method.

## A.2 PROOF OF PROPOSITION 2

*Proof.* We first prove the following lemma which connects the backward pass between two frameworks in the degenerate case.

**Lemma 6.** *Assume $Q_{\boldsymbol{ux}}^t = \boldsymbol{0}$ at all stages, then we have*

$$V_{\boldsymbol{x}}^t = \nabla_{\boldsymbol{x}_t} J, \; and \quad V_{\boldsymbol{xx}}^t = \nabla_{\boldsymbol{x}_t}^2 J, \quad \forall t. \tag{25}$$

*Proof.* It is obvious to see that Eq. 25 holds at $t = T$. Now, assume the relation holds at $t + 1$ and observe that at the time $t$, the backward passes take the form of

$$V_{\boldsymbol{x}}^t = Q_{\boldsymbol{x}}^t - Q_{\boldsymbol{ux}}^{t\mathsf{T}} (Q_{\boldsymbol{uu}}^t)^{-1} Q_{\boldsymbol{u}}^t = \ell_{\boldsymbol{x}}^t + f_{\boldsymbol{x}}^{t\mathsf{T}} \nabla_{\boldsymbol{x}_{t+1}} J = \nabla_{\boldsymbol{x}_t} J,$$

$$V_{\boldsymbol{xx}}^t = Q_{\boldsymbol{xx}}^t - Q_{\boldsymbol{ux}}^{t\mathsf{T}} (Q_{\boldsymbol{uu}}^t)^{-1} Q_{\boldsymbol{ux}}^t = \nabla_{\boldsymbol{x}_t} \{ \ell_{\boldsymbol{x}}^t + f_{\boldsymbol{x}}^{t\mathsf{T}} \nabla_{\boldsymbol{x}_{t+1}} J \} = \nabla_{\boldsymbol{x}_t}^2 J,$$

where we recall $J_t = \ell_t + J_{t+1}(f_t)$ in Eq. 8. $\qquad\square$

Now, Eq. 11 follows by substituting Eq. 25 to the definition of $Q_{\boldsymbol{u}}^t$ and $Q_{\boldsymbol{uu}}^t$

$$Q_{\boldsymbol{u}}^t = \ell_{\boldsymbol{u}}^t + f_{\boldsymbol{u}}^{t\mathsf{T}} V_{\boldsymbol{x}}^{t+1} = \ell_{\boldsymbol{u}}^t + f_{\boldsymbol{u}}^{t\mathsf{T}} \nabla_{\boldsymbol{x}_{t+1}} J = \nabla_{\boldsymbol{u}_t} J,$$

$$Q_{\boldsymbol{uu}}^t = \ell_{\boldsymbol{uu}}^t + f_{\boldsymbol{u}}^{t\mathsf{T}} V_{\boldsymbol{xx}}^{t+1} f_{\boldsymbol{u}}^t + V_{\boldsymbol{x}}^{t+1} \cdot f_{\boldsymbol{uu}}^t$$

$$= \ell_{\boldsymbol{uu}}^t + f_{\boldsymbol{u}}^{t\mathsf{T}} (\nabla_{\boldsymbol{x}_{t+1}}^2 J) f_{\boldsymbol{u}}^t + \nabla_{\boldsymbol{x}_{t+1}} J \cdot f_{\boldsymbol{uu}}^t$$

$$= \nabla_{\boldsymbol{u}_t} \{ \ell_{\boldsymbol{u}}^t + f_{\boldsymbol{u}}^{t\mathsf{T}} \nabla_{\boldsymbol{x}_{t+1}} J \} = \nabla_{\boldsymbol{u}_t}^2 J.$$

Consequently, the DDP feedback policy degenerates to layer-wise Newton update. $\qquad\square$

## A.3 PROOF OF PROPOSITION 3

*Proof.* We will prove Proposition 3 by backward induction. Suppose at layer $t + 1$, we have $V_{\boldsymbol{xx}}^{t+1} = \boldsymbol{z}_{\boldsymbol{x}}^{t+1} \otimes \boldsymbol{z}_{\boldsymbol{x}}^{t+1}$ and $\ell_t \equiv \ell_t(\boldsymbol{u}_t)$, then Eq. 3 becomes

$$Q_{\boldsymbol{xx}}^t = f_{\boldsymbol{x}}^{t\mathsf{T}} V_{\boldsymbol{xx}}^{t+1} f_{\boldsymbol{x}}^t = f_{\boldsymbol{x}}^{t\mathsf{T}} (\boldsymbol{z}_{\boldsymbol{x}}^{t+1} \otimes \boldsymbol{z}_{\boldsymbol{x}}^{t+1}) f_{\boldsymbol{x}}^t = (f_{\boldsymbol{x}}^{t\mathsf{T}} \boldsymbol{z}_{\boldsymbol{x}}^{t+1}) \otimes (f_{\boldsymbol{x}}^{t\mathsf{T}} \boldsymbol{z}_{\boldsymbol{x}}^{t+1})$$

$$Q_{\boldsymbol{ux}}^t = f_{\boldsymbol{u}}^{t\mathsf{T}} V_{\boldsymbol{xx}}^{t+1} f_{\boldsymbol{x}}^t = f_{\boldsymbol{u}}^{t\mathsf{T}} (\boldsymbol{z}_{\boldsymbol{x}}^{t+1} \otimes \boldsymbol{z}_{\boldsymbol{x}}^{t+1}) f_{\boldsymbol{x}}^t = (f_{\boldsymbol{u}}^{t\mathsf{T}} \boldsymbol{z}_{\boldsymbol{x}}^{t+1}) \otimes (f_{\boldsymbol{x}}^{t\mathsf{T}} \boldsymbol{z}_{\boldsymbol{x}}^{t+1}).$$

Setting $\boldsymbol{q}_{\boldsymbol{x}}^t := f_{\boldsymbol{x}}^{t\mathsf{T}} \boldsymbol{z}_{\boldsymbol{x}}^{t+1}$ and $\boldsymbol{q}_{\boldsymbol{u}}^t := f_{\boldsymbol{u}}^{t\mathsf{T}} \boldsymbol{z}_{\boldsymbol{x}}^{t+1}$ will give the first part of Proposition 3.

Next, to show the same factorization structure preserves through the preceding layer, it is sufficient to show $V_{\boldsymbol{xx}}^t = \boldsymbol{z}_{\boldsymbol{x}}^t \otimes \boldsymbol{z}_{\boldsymbol{x}}^t$ for some vector $\boldsymbol{z}_{\boldsymbol{x}}^t$. This is indeed the case.

$$V_{\boldsymbol{xx}}^t = Q_{\boldsymbol{xx}}^t - Q_{\boldsymbol{ux}}^{t\mathsf{T}} (Q_{\boldsymbol{uu}}^t)^{-1} Q_{\boldsymbol{ux}}^t$$

$$= \boldsymbol{q}_{\boldsymbol{x}}^t \otimes \boldsymbol{q}_{\boldsymbol{x}}^t - (\boldsymbol{q}_{\boldsymbol{u}}^t \otimes \boldsymbol{q}_{\boldsymbol{x}}^t)^{\mathsf{T}} (Q_{\boldsymbol{uu}}^t)^{-1} (\boldsymbol{q}_{\boldsymbol{u}}^t \otimes \boldsymbol{q}_{\boldsymbol{x}}^t)$$

$$= \boldsymbol{q}_{\boldsymbol{x}}^t \otimes \boldsymbol{q}_{\boldsymbol{x}}^t - (\boldsymbol{q}_{\boldsymbol{u}}^{t\mathsf{T}} (Q_{\boldsymbol{uu}}^t)^{-1} \boldsymbol{q}_{\boldsymbol{u}}^t)(\boldsymbol{q}_{\boldsymbol{x}}^t \otimes \boldsymbol{q}_{\boldsymbol{x}}^t),$$

where the last equality follows by observing $\boldsymbol{q}_{\boldsymbol{u}}^{t\mathsf{T}} (Q_{\boldsymbol{uu}}^t)^{-1} \boldsymbol{q}_{\boldsymbol{u}}^t$ is a scalar.

Set $\boldsymbol{z}_{\boldsymbol{x}}^t = \sqrt{1 - \boldsymbol{q}_{\boldsymbol{u}}^{t\mathsf{T}} (Q_{\boldsymbol{uu}}^t)^{-1} \boldsymbol{q}_{\boldsymbol{u}}^t} \; \boldsymbol{q}_{\boldsymbol{x}}^t$ will give the desired factorization.

$\qquad\square$

## A.4 DERIVATION OF EQ. 12

For notational simplicity, we drop the superscript $t$ and denote $V'_{x'} \triangleq \nabla_x V_{t+1}(x_{t+1})$ as the derivative of the value function at the next state.

$$Q_u = \ell_u + f_u^\mathsf{T} V'_{x'} = \ell_u + g_u^\mathsf{T} \sigma_h^\mathsf{T} V'_{x'},$$

$$Q_{uu} = \ell_{uu} + \frac{\partial}{\partial u} \{g_u^\mathsf{T} \sigma_h^\mathsf{T} V'_{x'}\}$$

$$= \ell_{uu} + g_u^\mathsf{T} \sigma_h^\mathsf{T} \frac{\partial}{\partial u}\{V'_{x'}\} + g_u^\mathsf{T}(\frac{\partial}{\partial u}\{\sigma_h\})^\mathsf{T} V'_{x'} + (\frac{\partial}{\partial u}\{g_u\})^\mathsf{T} \sigma_h^\mathsf{T} V'_{x'}$$

$$= \ell_{uu} + g_u^\mathsf{T} \sigma_h^\mathsf{T} V'_{x'x'} \sigma_h g_u + g_u^\mathsf{T}(V'^\mathsf{T}_{x'} \sigma_{hh} g_u) + g_{uu}^\mathsf{T} \sigma_h^\mathsf{T} V'_{x'}$$

$$= \ell_{uu} + g_u^\mathsf{T}(V_{hh} + V'_{x'} \cdot \sigma_{hh})g_u + V_h \cdot g_{uu}$$

The last equation follows by recalling $V_h \triangleq \sigma_h^\mathsf{T} V'_{x'}$ and $V_{hh} \triangleq \sigma_h^\mathsf{T} V'_{x'x'} \sigma_h$. Follow similar derivation, we have

$$Q_x = \ell_x + g_x^\mathsf{T} V_h$$
$$Q_{xx} = \ell_{xx} + g_x^\mathsf{T}(V_{hh} + V'_{x'} \cdot \sigma_{hh})g_x + V_h \cdot g_{xx} \tag{26}$$
$$Q_{ux} = \ell_{ux} + g_u^\mathsf{T}(V_{hh} + V'_{x'} \cdot \sigma_{hh})g_x + V_h \cdot g_{ux}$$

**Remarks.** For feedforward networks, the computational overhead in Eq. 12 and 26 can be mitigated by leveraging its affine structure. Since $g$ is bilinear in $x_t$ and $u_t$, the terms $g_{xx}^t$ and $g_{uu}^t$ vanish. The tensor $g_{ux}^t$ admits a sparse structure, whose computation can be simplified to

$$[g_{ux}^t]_{(i,j,k)} = 1 \quad \text{iff} \quad j = (k-1)n_{t+1} + i,$$
$$[V_h^t \cdot g_{ux}^t]_{((k-1)n_{t+1}:kn_{t+1},k)} = V_h^t. \tag{27}$$

For the coordinate-wise nonlinear transform, $\sigma_h^t$ and $\sigma_{hh}^t$ are diagonal matrix and tensor. In most learning instances, stage-wise losses typically involved with weight decay alone; thus the terms $\ell_x^t, \ell_{xx}^t, \ell_{ux}^t$ also vanish.

## A.5 DERIVATION OF EQ. 18

Eq. 18 follows by an observation that the feedback policy $\mathbf{K}_t \delta x_t = -(Q_{uu}^t)^{-1} Q_{ux}^t \delta x_t$ stands as the minimizer of the following objective

$$\mathbf{K}_t \delta x_t = \underset{\delta u_t(\delta x_t) \in \Gamma'(\delta x_t)}{\arg\min} \|\nabla_{u_t} Q(x_t + \delta x_t, u_t + \delta u_t(\delta x_t)) - \nabla_{u_t} Q(x_t, u_t)\|, \tag{28}$$

where $\Gamma'(\delta x_t)$ denotes all affine mappings from $\delta x_t$ to $\delta u_t$ and $\|\cdot\|$ can be any proper norm in the Euclidean space. Eq. 28 follows by the Taylor expansion of $Q(x_t + \delta x_t, u_t + \delta u_t)$ to its first order,

$$\nabla_{u_t} Q(x_t + \delta x_t, u_t + \delta u_t) = \nabla_{u_t} Q(x_t, u_t) + Q_{ux}^t \delta x_t + Q_{uu}^t \delta u_t.$$

When $Q = J$, we will arrive at Eq. 18. From Proposition 2, we know the equality holds when all $Q_{xu}^s$ vanish for $s > t$. In other words, the approximation in Eq. 18 becomes equality when all aferward layer-wise objectives $s > t$ are expanded only w.r.t. $u_s$.

## A.6 EXPERIMENT DETAIL

### A.6.1 SETUP

**Clarification Dataset.** All networks in the classification experiments are composed of 5-6 layers. For the intermediate layers, we use ReLU activation on all dataset, except Tanh on WINE and DIGITS. We use identity mapping at the last prediction layer on all dataset except WINE, where we use sigmoid instead to help distinguish the performance among optimizers. For feedforward networks, the dimension of the hidden state is set to 10-32. On the other hand, we use standard $3 \times 3$ convolution kernels for all CNNs. The batch size is set 8-32 for dataset trained with feedforward networks,

Table 5: Hyper-parameter search

| Methods | Learning Rate |
|---|---|
| SGD | $(7e\text{-}2, 5e\text{-}1)$ |
| Adam & RMSprop | $(7e\text{-}4, 1e\text{-}2)$ |
| EKFAC | $(1e\text{-}2, 3e\text{-}1)$ |

and 128 for dataset trained with convolution networks. For each baseline we select its own hyper-parameter from an appropriate search space, which we detail in Table 5. We use the implementation in https://github.com/Thrandis/EKFAC-pytorch for EKFAC and implement our own E-MSA in PyTorch since the official code released from Li et al. (2017) does not support GPU implementation. We impose the GN factorization presented in Proposition 3 for all CNN training. Regarding the machine information, we conduct our experiments on GTX 1080 TI, RTX TITAN, and four Tesla V100 SXM2 16GB.

**Procedure to Generate Fig. 5.** First, we perform standard DDPNOpt steps to compute layer-wise policies. Next, we conduct singular-value decomposition on the feedback matrix $(\boldsymbol{k}_t, \boldsymbol{K}_t)$. In this way, the leading right-singular vector corresponding to the dominating that the feedback policy shall respond with. Since this vector is with the same dimension as the hidden state, which is most likely not the same as the image space, we project the vector back to image space using the techniques proposed in (Zeiler & Fergus, 2014). The pseudo code and computation diagram are included in Alg. 4 and Fig. 7.

---

**Algorithm 4** Visualizing the Feedback Policies

1: **Input:** Image $\boldsymbol{x}$ (we drop the time subscript for notational simplicity, *i.e.* $\boldsymbol{x} \equiv \boldsymbol{x}_0$)
2: Perform backward pass of DDPNOpt. Compute $(\boldsymbol{k}_t, \boldsymbol{K}_t)$ backward
3: Perform SVD on $\boldsymbol{K}_t$
4: Extract the right-singular vector corresponding to the largest singular value, denoted $v_{\max} \in \mathbb{R}^{n_t}$
5: Project $v_{\max}$ back to the image space using deconvolution procedures introduced in (Zeiler & Fergus, 2014)

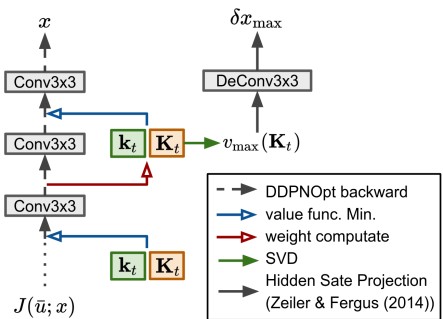

Figure 7: Pictorial illustration for Alg. 4.

### A.6.2 ADDITIONAL EXPERIMENT AND DISCUSSION

**Batch trajectory optimization on synthetic dataset.** One of the difference between DNN training and trajectory optimization is that for the former, we aim to find an ultimate control law that can drive every data point in the training set, or sampled batch, to its designed target. Despite seemly trivial from the ML perspective, this is a distinct formulation to OCP since the optimal policy typically varies at different initial state. As such, we validate performance of DDPNOpt in batch trajectories optimization on a synthetic dataset, where we sample data from $k \in \{5, 8, 12, 15\}$ Gaussian clusters in $\mathbb{R}^{30}$. Since conceptually a DNN classifier can be thought of as a dynamical system guiding trajectories of samples toward the target regions belong to their classes, we hypothesize that for the DDPNOpt to show its effectiveness on batch training, the feedback policy must act as an ensemble policy that combines the locally optimal policy of each class. Fig. 8 shows the spectrum distribution, sorted in a descending order, of the feedback policy in the prediction layer. The result shows that the number of nontrivial eigenvalues matches exactly the number of classes in each setup (indicated by the vertical dashed line). As the distribution in the prediction layer concentrates to $k$ bulks through training, the eigenvalues also increase, providing stronger feedback to the weight update.

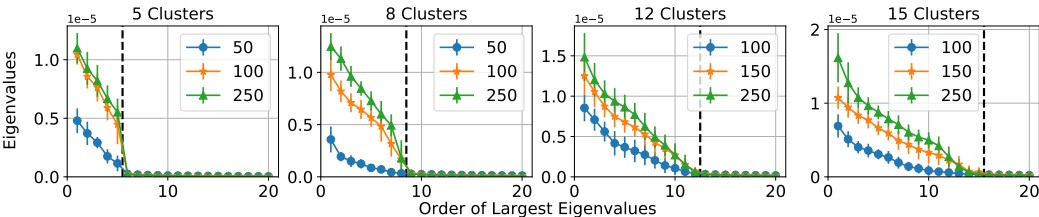

Figure 8: Spectrum distribution on synthetic dataset.

**Ablation analysis on Adam.** Fig. 9 reports the ablation analysis on Adam using the same setup as in Fig. 4a, *i.e.* we keep all hyper-parameters the same for each experiment so that the performance

difference only comes from the existence of feedback policies. It is clear that the improvements from the feedback policies remain consistent for Adam optimizer.

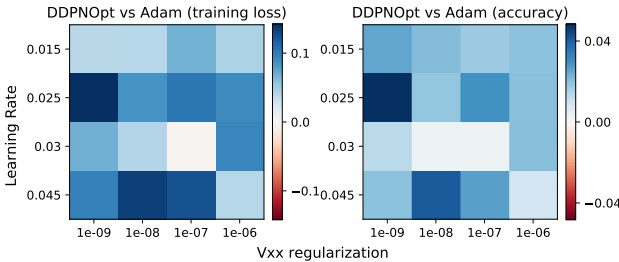

Figure 9: Additional experiment for Fig. 4a where we compare the performance difference between DDPNOpt and Adam. Again, all grids report values averaged over 10 random seeds.

**Ablation analysis on DIGITS compared with best-tuned baselines.** Fig. 4 reports the performance difference between baselines and DDPNOpt under different hyperparameter setups. Here, we report the numerical values when each baseline uses its best-tuned learning rate (which is the values we report in Table 3) and compare with its DDPNOpt counterpart using the same learning rate. As shown in Tables 6, 7, and 8, for most cases extending the baseline to accept the Bellman framework improves the performance.

Table 6: Learning rate $= 0.1$

|  | SGD | DDPNOpt with $M_t = I_t$ |
|---|---|---|
| Train Loss | 0.035 | **0.032** |
| Accuracy (%) | 95.36 | **95.52** |

Table 7: Learning rate $= 0.001$

|  | RMSprop | DDPNOpt with $M_t = \mathrm{diag}(\sqrt{\mathbb{E}[Q_u^t \odot Q_u^t]} + \epsilon)$ |
|---|---|---|
| Train Loss | 0.058 | **0.052** |
| Accuracy (%) | 94.33 | **94.63** |

Table 8: Learning rate $= 0.03$

|  | EKFAC | DDPNOpt with $M_t = \mathbb{E}[x_t x_t^\mathsf{T}] \otimes \mathbb{E}[V_h^t V_h^{t\,\mathsf{T}}]$ |
|---|---|---|
| Train Loss | 0.074 | **0.067** |
| Accuracy (%) | **95.24** | 95.19 |

**Numerical absolute values in ablation analysis (DIGITS).** Fig. 4a reports the *relative* performance between each baseline and its DDPNOpt counterpart under different learning rate and regularization setups. In Table 9 and 10, we report the *absolute* numerical values of this experiment. For instance, the most left-upper grid in Fig. 4a, *i.e.* the training loss difference between DDPNOpt and SGD with learning rate 0.4 and $V_{xx}$ regularization $5 \times 10^{-5}$, corresponds to $0.1974 - 0.1662$ in Table 9. All values in these tables are averaged over 10 seeds.

Table 9: Training Loss. ($\epsilon_{V_{xx}}$ denotes the Tikhonov regularization on $V_{xx}$.)

|  |  | SGD | DDPNOpt with $M_t = I_t$ | | | |
|---|---|---|---|---|---|---|
|  |  |  | $\epsilon_{V_{xx}} = 5 \times 10^{-5}$ | $1 \times 10^{-4}$ | $5 \times 10^{-4}$ | $1 \times 10^{-3}$ |
| Learn Rate | 0.4 | 0.1974 | 0.1662 | 0.1444 | 0.1322 | **0.1067** |
|  | 0.6 | 0.5809 | 0.4989 | 0.4867 | 0.3263 | **0.2764** |
|  | 0.7 | 1.0493 | 0.9034 | 0.8240 | 0.6592 | **0.5381** |
|  | 0.8 | 1.7801 | 1.6898 | 1.4597 | **1.1784** | 1.3166 |

| | RMSprop | DDPNOpt with $M_t = \text{diag}(\sqrt{\mathbb{E}[Q_u^t \odot Q_u^t]} + \epsilon)$ | | | |
| --- | --- | --- | --- | --- | --- |
| | | $\epsilon_{V_{xx}} = 1 \times 10^{-9}$ | $1 \times 10^{-8}$ | $5 \times 10^{-6}$ | $1 \times 10^{-5}$ |
| Learn Rate 0.01 | 0.1949 | 0.1638 | 0.1714 | 0.1746 | **0.1588** |
| 0.02 | 0.4691 | 0.4559 | 0.4489 | **0.4390** | 0.4773 |
| 0.03 | 0.8156 | **0.7675** | 0.7736 | 0.7790 | 0.7893 |
| 0.045 | 1.3103 | 1.2740 | 1.2956 | **1.2568** | 1.2758 |

| | EKFAC | DDPNOpt with $M_t = \mathbb{E}[x_t x_t^\top] \otimes \mathbb{E}[V_h^t V_h^{t\top}]$ | | | |
| --- | --- | --- | --- | --- | --- |
| | | $\epsilon_{V_{xx}} = 1 \times 10^{-7}$ | $5 \times 10^{-7}$ | $5 \times 10^{-6}$ | $1 \times 10^{-5}$ |
| Learn Rate 0.05 | 0.0757 | **0.0636** | 0.0659 | 0.0691 | 0.0717 |
| 0.09 | 0.2274 | **0.2087** | 0.2164 | 0.2091 | 0.2223 |
| 0.1 | 0.3260 | 0.2771 | 0.3003 | 0.2543 | **0.2510** |
| 0.3 | 0.5959 | 0.5462 | **0.5282** | 0.5299 | 0.5858 |

Table 10: Accuracy (%). ($\epsilon_{V_{xx}}$ denotes the $V_{xx}$ regularization.)

| | SGD | DDPNOpt with $M_t = I_t$ | | | |
| --- | --- | --- | --- | --- | --- |
| | | $\epsilon_{V_{xx}} = 5 \times 10^{-5}$ | $1 \times 10^{-4}$ | $5 \times 10^{-4}$ | $1 \times 10^{-3}$ |
| Learn Rate 0.4 | 91.46 | 91.98 | 92.71 | 92.90 | **93.12** |
| 0.6 | 81.73 | 83.64 | 84.09 | 88.39 | **89.39** |
| 0.7 | 70.48 | 73.42 | 75.44 | 80.62 | **82.87** |
| 0.8 | 55.76 | 57.70 | 62.82 | **70.23** | 65.01 |

| | RMSprop | DDPNOpt with $M_t = \text{diag}(\sqrt{\mathbb{E}[Q_u^t \odot Q_u^t]} + \epsilon)$ | | | |
| --- | --- | --- | --- | --- | --- |
| | | $\epsilon_{V_{xx}} = 1 \times 10^{-9}$ | $1 \times 10^{-8}$ | $5 \times 10^{-6}$ | $1 \times 10^{-5}$ |
| Learn Rate 0.01 | 91.48 | 92.14 | 91.80 | 91.73 | **92.52** |
| 0.02 | 84.15 | 84.82 | 85.02 | **85.23** | 83.00 |
| 0.03 | 73.07 | 75.24 | **75.73** | 74.29 | 74.16 |
| 0.045 | 59.80 | 59.16 | 60.98 | **61.75** | 59.87 |

| | EKFAC | DDPNOpt with $M_t = \mathbb{E}[x_t x_t^\top] \otimes \mathbb{E}[V_h^t V_h^{t\top}]$ | | | |
| --- | --- | --- | --- | --- | --- |
| | | $\epsilon_{V_{xx}} = 1 \times 10^{-7}$ | $5 \times 10^{-7}$ | $5 \times 10^{-6}$ | $1 \times 10^{-5}$ |
| Learn Rate 0.05 | 93.70 | 93.84 | 93.88 | **94.31** | 94.06 |
| 0.09 | 90.84 | 91.13 | **91.45** | 91.23 | 91.24 |
| 0.1 | 88.88 | 89.69 | 89.89 | 90.18 | **90.94** |
| 0.3 | 81.82 | 83.79 | 84.09 | **84.15** | 82.55 |

**More experiments on vanishing gradient.** Recall that Fig. 6 reports the training performance using MMC loss on Sigmoid-activated networks. In Fig. 10a, we report the result when training the same networks but using CE loss (notice the numerical differences in the $y$ axis for different objectives). None of the presented optimizers were able to escape from vanishing gradient, as evidenced by the vanishing update magnitude. On the other hands, changing the networks to ReLU-activated networks eliminates the vanishing gradient, as shown in Fig. 10b.

Fig. 11 reports the performance with other first-order adaptive optimizers including Adam and RMSprop. In general, adaptive first-order optimizers are more likely to escape from vanishing gradient since the diagonal precondition matrix (recall $M_t = \mathbb{E}[J_{u_t} \odot J_{u_t}]$ in Table 2) rescales the vanishing update to a fixed norm. However, as shown in Fig. 11, DDPNOpt* (the variant of DDPNOpt that utilizes similar adaptive first-order precondition matrix) converges faster compared with these adaptive baselines.

Fig. 12 illustrates the selecting process on the learing-rate tuning when we report Fig. 6. The training performance for both SGD-VGR and EKFAC remains unchanged when tuning the learning rate. In

practice, we observe unstable training with SGD-VGR when the learing rate goes too large. On the other hands, DDPNOpt and DDPNOpt2nd are able to escape from VG with all tested learning rates. Hence, Fig. 6 combines Fig. 12a (SGD-VGR-lr0.1) and Fig. 12c (EKFAC-lr0.03, DDPNOpt-lr0.03, DDPNOpt2nd-lr0.03) for best visualization.

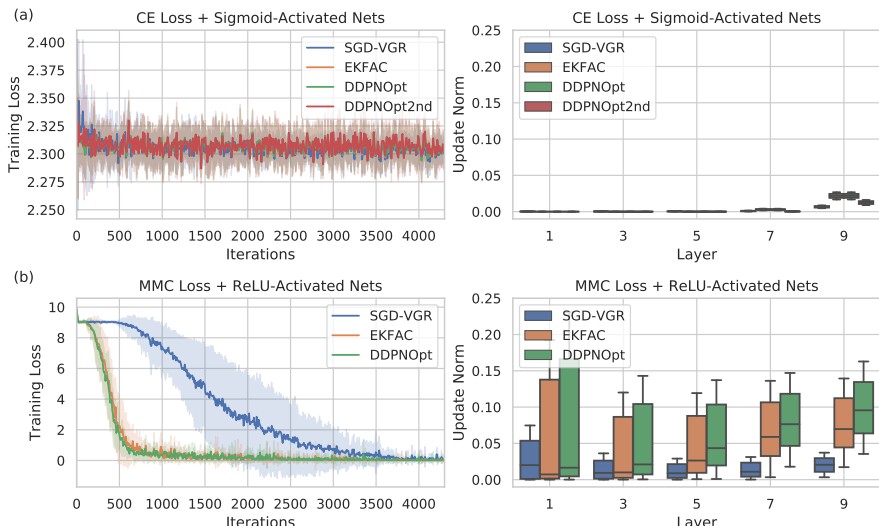

Figure 10: Vanishing gradient experiment for different losses and nonlinear activation functions.

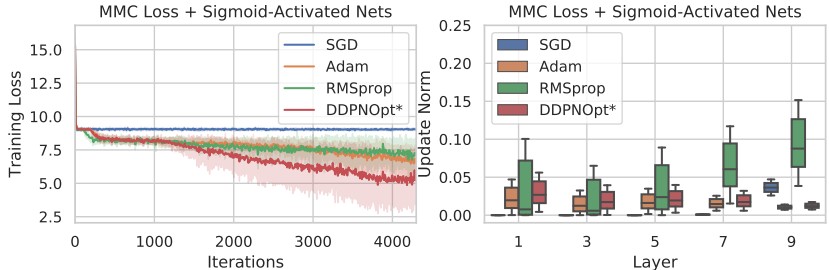

Figure 11: Vanishing gradient experiment for other optimizers. The legend "DDPNOpt*" denotes DDPNOpt with adaptive diagonal matrix.

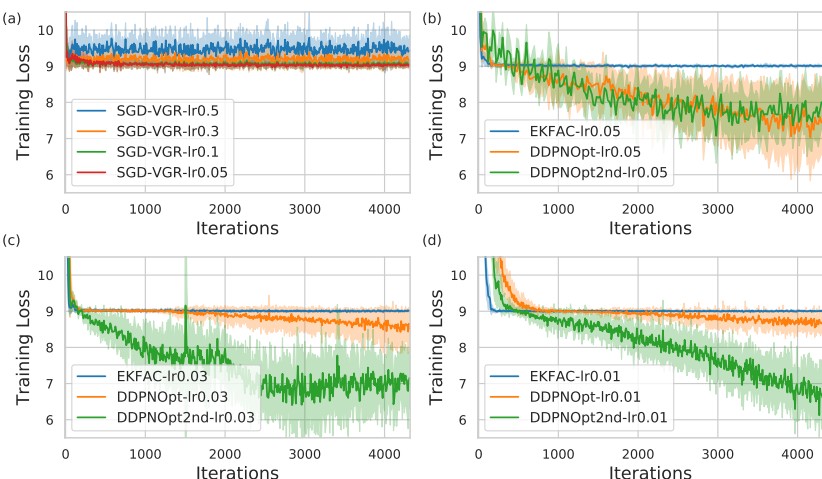

Figure 12: Vanishing gradient experiment for different learning rate setups.

