# OpenReview forum: "DDPNOpt: Differential Dynamic Programming Neural Optimizer"
_ICLR.cc/2021/Conference — ICLR 2021 Spotlight_

### Official Review · AnonReviewer4 · 2020-10-29
**A novel optimal control based optimizer with interesting connections to existing methods**

**Rating:** 7
**Confidence:** 3

**Review:**


A. Summary:
This paper proposes a second-order optimizer, DDP-NOpt, for neural network training. This optimizer views neural networks as a discrete-time non-linear dynamic system and derives an efficient optimizer from differential dynamic programming.
Besides, this paper also connects the existing back-propagation with gradient descent to the DDP framework and explains why the proposed optimizer is better than others under the DDP framework.
Finally, the experiments demonstrate that the proposed method is superior to the other optimal control inspired method and second-order method, as well as competitive with the standard first-order method.

B. Strength:
1. This paper gives a clear discussion about the difference between the proposed method and previous works, which motivates the author to explore the optimizer in the line of approximate dynamic programming, specifically differential dynamic programming.
2. The paper gives sufficient background to help the readers to understand the idea of viewing networks as non-linear dynamic systems and optimize it with differential dynamic programming. It also establishes the connection between existing first-order and second-order optimizer to the DDP framework and gives a clear explanation about why the proposed method is superior in section.3.
3. The experiments are comprehensive. Besides the model accuracy, it also compares the sensitivity to hyperparameters of different optimizers, which is intuitive via visualization. Training a relatively deep network with sigmoid activation also proves the superiority of the algorithm over existing methods.

C. Weakness:
1. There are two sets of terminology through this paper, one is for optimal control, and the other is for deep learning, which is more familiar to me. When reading the section.4, I have to turn back to the section.2 frequently to double confirm the terms. So is that possible to introduce the DDPNOpt right after the DDP, when the reader's memory about the DDP is still fresh? After the DDPNOpt, then we can discuss the connection between BP+GD with DDP, as well as the relation of DDPNOpt with other optimizers.
2. The network structures used in the experiments are not clarified. Different Network structures lead to different dynamics and objective landscape, which may affect the relative effectiveness of the proposed optimizer. For example, it is well-known that networks with residual connection (resnet, densenet) are easier to optimize than the ones without residual connections.  So it would be good if the author could clarify the specific network structure they are experimenting with and demonstrate the generality of the optimizer on different structures.
4. Does the SGD in the experiment have momentum? If not, it is better to compare with SGD+momentum since it is a more popular choice than vanilla SGD.
5. The analysis in the paper is based on the full-batch case. How the relative performance varies according to the batch size? Only runtime comparison contains the results for different batch sizes.

D. Justification of the score:
I am not an expert on this topic, so I made the voting from the perspective of a broader audience of the ICLR community. In general, it is a good paper in terms of the novelty and the algorithm. There are some paper organization issues making the paper a little bit difficult to follow, but these issues are fixable, and the paper will be more friendly to a broader audience after this.

E. Expectation for the rebuttal:
I hope the authors could address my questions in C.weakness and make the paper easier to follow for a broader audience.

---

> ### Author Response · Authors · 2020-11-18
> **Author Response to Reviewer 4**
>
> We thank the reviewer for the valuable feedback. We have re-organized the presentation with several amendments in the revised version (changes are marked with blue). Clarifications are attached below.
>
> **1. Organization**
> - We thank the reviewer for the valuable suggestion and we have revised the paper organization. In the new version, we introduce DDPNOpt right after we present DDP and its connection to existing methods, so that the derivation of DDPNOpt is close to DDP. We hope this revision makes it easier when comparing between these three frameworks (DDP, DDPNOpt, BP). Discussion on _'Role of the feedback $K_t$ and $\delta x_t$'_ (the latter part of Sec 3 in the original submission) now appears after the derivation of DDPNOpt. As suggested by Reviewers 1 and 2, presenting the connection between DDP and BP in the early text provides useful referencing point for the later derivation and the motivation of DDPNOpt. We hope this organization improves the reading experience for a broader audience.
>
> ****
>
> **2. Network architecture**
> - In this paper we focus on evaluating the performance of DDPNOpt with networks consisting of fully-connected and convolution layers, since the derivatives of the layer-wise objective ($Q_t$) can be efficiently evaluated on these layers (Eq 12). In Table 3, the feedforward networks consist of 5 fully-connected layers with the hidden dimension ranging from 10 to 32, depending on the size of dataset. The convolution network (CNN) consist of 4 convolution layers (with 3 by 3 kernel, 32 channels), followed by 2 fully-connected layers. All networks use ReLU activation, except Tanh for WINE and DIGITS, with affine mapping at the final prediction layer. Runtime comparison (Fig 3) reports the values with the same CNN model. Fig 4 and 6 are measured on DIGITS dataset; hence conducted with the same feedforward network. Finally, Fig 5 visualizes the feedback policies when training CNN. We include these discussions (network architecture and experiment setup) in the revised version (marked with blue in Sec 5).
> - When connecting other architectures, _e.g._ (overlapping) residual connections in ResNet and DenseNet, to the presenting optimal control perspective, the dynamical system interpretation (Eq 6) will require additional modifications. Since the dynamics is no longer a Markovian process, solving the Bellman equation (Eq 1) involves variables from earlier stages. While this can lead to very interesting formulation and will be an interesting direction to pursue in future works, it is out of the scope for the current submission.
>
> ****
>
> **3. SGD with momentum**
> - We thank the reviewer for pointing this out. In the revised version, Table 3 now reports the training performance of SGD tuned with momentum. We find an improvement in accuracy when training with CNN. However, we also note that DDPNOpt sustains the leading position when comparing with new baselines.
>
> ****
>
> **4. Performance on different batch size**
> - The experiments in Sec 5 cover the same range of batch size reported in Fig 3. In particular, Table 3 reports the training results of feedforward networks with the batch size ranging from 8 to 32, depending on the size of dataset. For CNN we set the batch size to 128. Fig 4 and 6 are run on the batch size respectively set to 8 and 20.
> - For each experiment, while the results may vary when changing the batch size, the relative performance between optimizers remains unchanged. Instead, we observe the key hyper-parameter that distinguishes DDPNOpt from baselines is the learning rate. This is because, from a trajectory optimization standpoint, DDPNOpt should best show its effectiveness on the training performance when the magnitude of the control update $\delta u_t$ becomes larger (since the state differential $\delta x_t$ increases, making and the feedback significant). In the application of DNN training, the magnitude of the control/weight update is directly affected by the learning rate. On the other hand, as shown in the Alg 3 (page 6), DDPNOpt averages the control update contributed from each sample among the batch data like other optimizers. Therefore, varying the batch size mostly only affect the computation (since the size of feedback policy grows wrt the batch size) rather than the relative training performance.

---

### Official Review · AnonReviewer3 · 2020-10-30
**Nice work!**

**Rating:** 7
**Confidence:** 4

**Review:**

This paper presents a novel alternative to SGD based on Differential Dynamic Programming that views deep networks as dynamical systems. I am very pleased with this contribution and I propose an accept. Some feedback below to further improve the work.

Positives:
1) The method seems to mitigate the effect of vanishing gradients
2) Learns models that can outperform state of the art optimizers as well as other methods based on the same family of approximate DP.
3) The method seems simple to implement and generalizes commonly used ones depending on the choice of Hessian approximation.
4) Despite approximations, the approach is more principled then most SOTA heuristics
5) The effect of state (hidden activations or pre-activations) feedback on vanishing is illustrated very nicely and experiments are neat.

Negatives:
1) Approximations come too much sparsely in the paper. Perhaps they should be stated clearly in the initial sections  - possibly not in full technical extent but to give an idea of the key elements that enable the technology to be used. A reader can get scared away by the amount of second order terms that you later discard. The regularization bit on V seems also quite fundamental and its role is often downplayed in papers about DDP. It should also be clear that these steps are needed to use the method.

2) It is not clear if in the end the method is still second order, considered the amount of approximation taken. This is perhaps a technical detail but it would be nice to clarify further when comparing to Newton for instance.

3) It is not clear which equations you implement in the final algorithm. In paticular, on page 6 just above Experiment there's a paragraph about `implementation details`  they do not seem to be details really. For instance, you discard some terms in eq (3) so it might be worth highlighting the ones you use directly there, if you don't want to rewrite the full equation.

My initial score is positive but I think this feedback should be accounted for the revised version and amendements are made to maintain this score.

---

> ### Author Response · Authors · 2020-11-18
> **Author Response to Reviewer 3**
>
> We thank the reviewer for the valuable suggestions. We have re-organized the presentation with several amendments in the revised version (changes are marked with blue). Clarifications are attached below.
>
> **1. Approximation and $V_{xx}$ regularization**
> - To highlight the role of approximation and $V_{xx}$ regularization, in Sec 1 we enumerate the key approximations and factorizations that enable the efficient usage of DDPNOpt. This includes _(i)_ curvature estimation from existing methods, _(ii)_ stabilization techniques used in trajectory optimization ($V_{xx}$ regularization), and _(iii)_ efficient factorization to OCP. Derivations on each component are now grouped in Sec 3.2 (titled 'Efficient Approximation and Factorization') to highlight its importance. We also add additional discussion regarding the effect of $V_{xx}$ regularization on both traditional trajectory optimization and DNN training (page 6). In particular, we highlight the role of $V_{xx}$ regularization in the application of DNN training: it prevents the value Hessian from low rank when the dimension of the hidden state varies during DDP backward pass; therefore provides stabilization for the training.
>
> ****
>
> **2. Second-order method**
> - Despite dropping the second-order terms on the dynamics, DDPNOpt still performs second-order expansions on the value function and propagates these Hessians ($V_{xx}$), together with the first-order derivatives, backward through the layers (Fig 1). This optimization scheme, _i.e._ computing layer-wise Hessian with linearizion on the layer dynamics, is closely related to the Gauss-Newton method [1], which can be further related to the Natural Gradient method such as KFAC (we refer the reviewer to Appendix C in [1]). Since both have been recognized as second order optimizers, we adopt the convention and identify DDPNOpt as second-order method. We note that from a standard convergence analysis standpoint, DDP exhibits similar locally quadratic convergence as Newton only when the dynamics are fully expanded to second order. For a pictorial illustration on the effect of dynamics expansion on the convergence behavior, we refer the reviewer to Fig 3 in [2]. We clarify this aspect in the revision when comparing DDP to Newton (page 7).
>
> ****
>
> **3. Implementation of the final algorithm**
> - The pseudo code of DDPNOpt is now attached on page 6 (Algorithm 3) right after we introduce DDPNOpt (Sec 3.2). The pseudo code links the computation at each stage to the corresponding equation in the previously-introduced derivation section. The 'implementation detail' is now replaced with a full paragraph discussing the effect of $V_{xx}$ regularization (page 6). Original discussion on first-order dynamics expansion is combined in Sec 3.2 when we discuss the derivatives of $Q_t$ with layer dynamics. Finally, Eq 12 is updated with only first-order expansion on the dynamics, which is used in the final algorithm. We hope the changes will make it clear which equations are used in the final implementation.
>
> ****
>
> [1] Practical Gauss-Newton Optimisation for Deep Learning
> [2] Differential Dynamic Programming on Lie Groups: Derivation, Convergence Analysis and Numerical Results

---

> > ### Comment · AnonReviewer3 · 2020-11-23
> > **Thanks for the clarifications.**
> >
> > I thank the authors for addressing my concerns and applying the necessary amendments to the paper.
> > I am satisfied with the response as well as the updates.

---

### Official Review · AnonReviewer1 · 2020-10-30
**Interesting connections between trajectory optimization and training deep networks**

**Rating:** 8
**Confidence:** 2

**Review:**

This paper draws connections between training neural networks, and trajectory optimization via differentiable dynamic programming (DDP) from optimal control.

The central idea is to think about the propagation of inputs through a deep neural network as a dynamical system, where the inputs/activations are the signal being propagated, and the weights of the network are control inputs that influence the trajectory of the activations through the network. From this perspective, training the neural network is like trying to control the trajectory (hence, trajectory optimization).

The paper does a good job of presenting this connection, with helpful figures and text to aid the reader.

Given these connections, the paper then goes on to draw explicit connections between trajectory optimization using differentiable dynamic programming (DDP), and standard optimization algorithms for training neural networks (e.g. gradient descent or even approximate 2nd order methods such as KFAC).

The crux of the algorithm is similar to backpropagation in that it involves forward and backward passes, but different quantities are computed for the backward pass. While this seems like a straightforward application of known techniques from control theory to deep learning, the paper does a good job of highlighting similarities and differences to backprop.

The paper tests the proposed algorithm on a handful of classification tasks. My main concerns are with the experiments, I think they could be more thorough and more clearly show the purported benefits of the DDPNOpt algorithm.

First, the paper claims that the DDPNOpt algorithm is more robust. Robust to what? Stochastic gradients? Larger step sizes in the optimization algorithm? I think the paper could do a better job of stating how the DDPNOpt algorithm is more robust, and providence direct evidence demonstrating that. As far as I can tell, the only evidence presented for robustness is the toy illustration in Fig 2.

Second, as the DDPNOpt algorithm optimizes the same total objective as the baseline optimizers, I would expect them to (eventually) reach the same loss. Is this correct? If so, why report just the final accuracies in Table 3? It seems more pertinent to show the entire training trajectory for these problems.

Third, for the vanishing gradients experiment, it is hard to tell if the reason the other algorithms perform poorly is strictly due to vanishing gradients. How are the step sizes for each algorithm tuned? Are the other optimizers stuck at a saddle point? Does the stark difference in performance go away if one were to switch to using ReLU activations, which presumably do not suffer from vanishing gradients as much?

Overall, I think the connections drawn between optimal control and neural network training are themselves interesting and thought provoking, even with my caveats about the experiments.

---

> ### Author Response · Authors · 2020-11-18
> **Author Response to Reviewer 1**
>
> We thank the reviewer for the valuable feedback. We have made several amendments in the revised version (changes are marked with blue). Clarifications are attached below.
>
> **1. Robustness of DDPNOpt**
> - From a trajectory optimization standpoint, the DDP feedback policies robustify the optimization in the sense that they stabilize the control update by taking the state differentials $\delta x_t$ into account. As pointed out by the reviewer, it can be seen from the toy illustration in Fig 2 that the state differentials $\delta x_t$ becomes significant through propagation, or when the magnitude of the control updates $\delta u_t$ increases. In the application of DNN training, the magnitude of the control/weight update is directly affected by the learning rate. Therefore, one should expect DDPNOpt to show its robustness on training performance when this hyper-parameter becomes larger. This is verified empirically in Fig 4a, where we observe larger performance improvement from DDPNOpt as the learning rate increases. In short, DDPNOpt robustifies the training performance when a further step size, _i.e._ a larger control update, is taken. In the revision, we include some clarifications on this aspect in both Sec 1 and 5.
>
> ****
>
> **2. Convergence behaviors for different optimizer**
> - In general, different classes of optimizer exhibit distinct convergence behaviors and may not reach the same loss/accuracy (even for long-enough training), as the prevalence of (local) minima exists on the landscape of DNNs. In our experiments, we observe a similar phenomenon. In Tables 9 and 10 (Appendix A.7.4), we report the numerical values of loss/accuracy for each grid in Fig 4a. All values are computed after training converge. While EKFAC can achieve similar accuracy as SGD, their final training losses differ by an order of magnitude, and the accuracy for each optimizer also varies under different setups. Even when all optimizers are run under the best-tuned configurations (Table 3), the accuracies can vary up to 5-10%. Therefore, comparison of accuracy stands as an important evaluation metric for the initial step.
> - As for the comparison of the entire training trajectory, we present the case study in Fig 4b to highlight the performance difference wrt different baselines. The upper part of Fig 4b demonstrates DDPNOpt has stabler convergence in both loss/accuracy, while the bottom part showcases an example where the final training result looks similar (also indicated in Fig 4a) but DDPNOpt converges faster compared with the baseline. In practice, we find that the difference in convergence behavior tends to become obvious for larger learning rates. These values may differ from the best-tuned configuration reported in Table 3.
>
> ****
>
> **3. Vanishing gradient (VG) experiment**
> - In Fig 11,12,13 (Appendix A.7.2), we provide additional experiments on VG. Fig 11a reports the training result with the same Sigmoid network but using the CE loss (notice the numerical differences in the y-axis for different objectives). None of the presented optimizers were able to escape from VG, as evidenced by the degenerate update magnitude.
> - On the other hand, changing the activation to ReLU eliminates the VG (Fig 11b), as conjectured by the reviewer. While SGD-VGR, EKFAC, and DDPNOpt can all escape from VG with ReLU network, DDPNOpt admits slightly faster convergence with smaller variance compared with the baselines. We are not able to obtain the results for DDPNOpt2nd since ReLU activation is unbounded and has degenerate second-order derivative. These properties make the training unstable and often degrade the performance. Investigations along this direction will be interesting future works.
> - Next, Fig 12 illustrates the selecting process on tuning the learning rate when we report Fig 6. For each baseline, we draw multiple learning rates from an optimizer-dependent search space, which we detail in Table 5. As shown in Fig 12, the training performance for both SGD-VGR and EKFAC remains unchanged when tuning the learning rate. In practice, we observe unstable training with SGD-VGR when the learning rate becomes larger. On the other hand, DDPNOpt and DDPNOpt2nd are able to escape from VG with all tested learning rates. Hence, Fig 6 combines Fig 12a (SGD-VGR-lr0.1) and Fig 12c for best visualization.
> - Finally, Fig 13 reports the performance with other first-order adaptive optimizers, such as Adam and RMSprop. In general, adaptive first-order optimizers are more likely to escape from VG since the diagonal precondition matrix (_i.e._ $M_t = diag(\sqrt{J_{u_t} \odot J_{u_t}})$ in Table 2) rescales the vanishing update to a fixed norm. However, DDPNOpt* (the variant of DDPNOpt that utilizes similar adaptive first-order precondition matrix) converges faster compared with these adaptive baselines.

---

> > ### Comment · AnonReviewer1 · 2020-11-25
> > **Thank you for the clarifications**
> >
> > Thanks to the authors for their clarifications, and for the additional experiments.

---

### Official Review · AnonReviewer2 · 2020-11-01

**Rating:** 7
**Confidence:** 3

**Review:**

Updated review
-----
-----

Given the unanimous support for acceptance amongst the reviewers, I don't think it is really necessary for me to provide a detailed update. The details of how the authors have addressed the concerns expressed in my initial review can be found in the follow up discussion. I now support acceptance without reservation.


Initial review
----
----

# Summary

The authors present a connection between differential dynamic programming (DDP) and back-propagation like gradient descent. The authors use this connection to derive an novel optimizer, DDPNOpt, which combines ideas from DDP to improve the performance with ideas from existing back-prop optimizers, e.g., ADAM, RMSprop, EKFAC, to make the overall approach more tractable. The authors show that this approach is competitive on several benchmarks with a reasonable runtime. Finally, the authors explore the behavior of its regularization term, as well as the effect of vanishing gradients on DDPNOpt.

# Reason for score

Overall, this seems like a good paper and a good first step in this direction. I have a several concerns about the experimental results which justify the lower score, but nothing that would necessarily be a deal-breaker. I would be happy to adjust my score if the authors could clarify some aspects of the experimental results.

# Pros

* The connection between gradient descent and DDP is interesting and, to the best of my knowledge, novel. The resulting approach of adjusting weight updates based on the differential of the hidden states is interesting. I would consider this work likely to generate interest in the community and to inspire follow up work. I'm curious to see if any connections could be drawn with ideas from convex/non-convex optimization in follow-up work.
* The ideas discussed flow well and are easy to follow. It's possible that a reader less familiar with optimal control would have a different experience but I found the arguments convincing with the necessary background concepts properly introduced.
* I found the presentation of alg. 1 and 2 to be insightful. It was useful to include them early in the text and provided something to reference when trying to absorb some of the later points. I caught myself going back to them whenever I had a doubt or question, enough to warrant an acknowledgement.

# Cons

* There are several aspects of the experimental results that raise some questions, but nothing that the authors wouldn't be able to address in their rebuttal. I will be adjusting my review based on the authors response.

# Questions for the authors

* Page 5, curvature approximation section, what does it mean for a layer $f^t$ to be "highly over-parameterized"? Do the authors use this to mean high-dimensional?
* Page 5, curvature approximation section, what is the dimensional of $Q^t_{uu}$? If it is quadratic in the parameters of a single layer, this doesn't seem intractable if materializing the full matrix is avoided, e.g., using a hessian-vector product formulation with a matrix-free linear solvers. Have the authors tried this?
* What are the standard deviations for the results in table 3?
* In figure 3, why does ADAM run faster when batch size is increased (up to 100)? Similarly, why is EKFAC's runtime mostly invariant to batch size? Were these also averaged over 10 runs? Some details about the implementation and how these were measured would help.
* What is the steepest slope for the memory usage of DDPNOpt in figure 3?
* How do ADAM, EKFAC and E-MSA keep a constant memory usage as the batch size increases?
* In table 4, where does the quadratic exponent of $X$ come from in ADAM's memory complexity? How does ADAM's memory complexity avoid a dependency on the batch size?
* Figure 4, what does the value of the $V_{xx}$ regularization axis correspond to?
* Figure 4a, how much of this is DDPNOpt doing better vs the comparison doing worse for certain learning rates? Can the authors show us what these plots would look like compared to the baseline using its best learning rate (or show two sets of plots one with the absolute performance of DDPNOpt and one with the absolute performance of the baseline)?
* Figure 4b, why use lr=0.01, Vxx=1e-5 for RMSprop instead of the seemingly favorable lr=0.045, Vxx=1e-9 (according to figure 4a)?
* Figure 5, it's not quite clear from the supporting text how exactly this figure was generated. Although I understand the high-level idea, I'm not sure I would know how to reproduce it exactly. Can the authors provide some more details concerning how the matrix whose eigenvector is being computed was constructed?
* Page 8, why change the loss function here? This makes it hard to relate these results with the ones discussed earlier. What do these results look like when using the same loss as what was used in table 3 (which I assume was CE)?
* How would the behavior of DDPNOpt (using 2nd order terms) be affected by using activations that have zero 2nd derivatives, e.g., ReLu?

# Minor comments and typos

* I think that, in its current state, figure 1 does more harm than good and could probably be omitted. This figure is dense in details and notation that isn't formally defined until much later in the paper which is likely to frustrate readers as they try to parse the figure while only half way through the introduction. Additionally, even now that I understand each of the terms involved, it's still not quite clear what the figure is meant to convey. Why does the weight update diagram not return new weights (or involve the weights at all)? Why are unused arrows faded in the weight updates for back prop but omitted from the backwards pass? Why does the backwards pass for DDPNOpt not show how all the necessary quantities needed for the weight updates are computed? The other side-by-side comparisons (alg. 1/2, table 2) do a good job at highlighting the similarities and differences between the different settings.
* Page 2, footnote, "given a time-dependent functional $\mathcal{F}_t(x_t, u_t)$ [...] as  $\nabla_x \mathcal{F}_t \equiv \mathcal{F}^t_x$ [...]", this is admittedly a  bit pedantic but as stated here, there are no variable $x$ to differentiate. I would recommend either using $\mathcal{F}_t(x, u)$, or $\nabla _{x_t} \mathcal{F}_t$
* Page 3, explicitly specifying where each derivative function is evaluated would help avoid confusion. Given the previous notation footnote, I would expect terms of the form $\mathcal{F}^t_x$ to be a function, but, as I understand it, they represent the derivatives evaluated at some point.
* Nitpick: how is the product between a vector and a 3D tensor defined? I know what the authors mean but not all readers might be comfortable with this notation. A few words mentioning it's a contraction on one dimension, or, preferably, a formal definition would help those readers (placing this in the appendix would be fine if space is limited).
* "by orders." This is used a few times without specifying what orders are being referred to. When discussing complexity, I have never seen this term used without being accompanied by "of magnitude". I don't believe it is correct to omit these words. If this is common in some literature that I am unfamiliar with, I would appreciate if the authors would let me know.
* There were a few other instances of typos and missing words which I now regret not writing down... In any case, it would be a good idea to carefully proof read the final draft before its eventual publication so that they can be found and corrected.

---

> ### Author Response · Authors · 2020-11-18
> **Author Response to Reviewer 2 (Part 1/3)**
>
> We thank the reviewer for the very helpful and detailed comments. We address all raised comments below. Changes are marked with blue in the revision.
>
> _**Experiment**_
>
> **1. Setup in runtime/memory comparison (Fig 3)**
> - Fig 3 reports the average values over 500 training iterations of 10 random trials. We run the experiment on standard GPU machines (please refer to Appendix A.6.1 for details) equipped with default GPU-accelerated library. The memory usage in general does not vary w.r.t. the randomness of seeds when other configurations (dataset, batch size, network architecture) are fixed. The numerical values of wall-clock time also remain nearly unchanged under the currently reported time scale (second). Hence, we drop the standard deviation for visualization simplicity.
>
> ****
>
> **2. Invariant runtime w.r.t. batch size for Adam/EKFAC (Fig 3)**
> - When the computation involved during training is highly parallelizable, modern GPU machines can accelerate the computation with parallel computing. For instance, during Back-propagation the first-order derivative is computed with repeated matrix multiplication over the same weight matrices for each sample among the batch data. So long as the GPU-accelerated library (cuDNN in our case) can fully utilize the GPU resource, the per-iteration runtime will remain relatively similar. As a result, for the range of batch size reported in Fig 3 ( $\le$ 300), the runtime of the baselines remains relatively similar. We note, however, that for a much larger batch size (>1024), the runtime for the baselines will start to increase as the GPU utilization starts to decay.
> - For the slight drop (~0.02 second) in the runtime of Adam near 64/128 batch size, we conjecture it may come from the fact that modern GPU-accelerated libraries perform optimization in memory allocation and data transmission among intermediate GPU processes [1]. For instance, running the same experiment with the batch size set to the power of 2 typically achieves faster per-iteration runtime. Therefore, the runtime may not necessarily grow regularly wrt the batch size. It can depend on the actual runtime processes or the implementation of the optimizers etc (as for Adam, we use the default implementation from pyTorch).
>
> ****
>
> **3. Memory complexity for Adam in Table 4**
> - The source of memory usage for each optimizer during the backward pass mainly consists of two components: (1) layer-wise update directions and (2) intermediate first (or second) order derivatives. Assume the dimensions of the state/control are $\mathbb{R}^n$, $\mathbb{R}^m$, and $B$ and $L$ are the batch size and network depth. Then for Adam (or any optimizer relies on BackProp), the memory taken in each components is (1)=$\mathcal{O}(mL)$, and (2)=$\mathcal{O}(Bn)$. The absence of $L$ in (2) is because standard automatic differentiation library (e.g. PyTorch) for DNN training is equipped with efficient memory manipulation. It keeps only minimal information along the computation graph. For instance, during Back-propagation, all proceeding first-order derivatives $\nabla_{x_s}J, s>t+1$ are discarded immediately once $\nabla_{x_t}J$ is computed. Now, for the type of networks (feedforward layers) presented in the paper, we have $m \approx n^2$. Also, $n$ is usually larger than $B$. Therefore, (1) will dominate. Keeping only $\mathcal{O}(mL)$ and noticing $m  \approx n^2$ give the reported complexity. We update the description of Table 4 in the revision to avoid confusion.
>
> ****
>
> **4. Memory comparison between DDPNOpt and baselines (Fig 3)**
> - Following the previous discussion, the primary memory usage in DDPNOpt backward pass comes from layer-wise feedback policies. With the same notation, the size of $\forall t, K_t$ grows in the order of $\mathcal{O}(BLnm)$. As a result, the memory usage for both DDPNOpt and DDP scales with the batch size. This is in contract to the baselines whose update directions only need $\mathcal{O}(mL)$ memory. Furthermore, when the batch size is comparatively small, the memory usage in the baselines may be dominated by other factors (e.g. the memory required for loading the network to the GPU), which are independent of the batch size.
> - We note that Table 4 reports only the algorithmic complexity during backward pass, whereas Fig 3 reports the actual training performance, which involves other factors such as parallel acceleration from the hardware system and memory consumed by  other stages. We have included some clarification in Sec 5, and we thank the reviewer for pointing out the confusion.
>
> ****
>
> **5. Standard deviation in Table 3**
> - We report the standard deviation in Table 11. The standard deviation is typically <0.2%, except for some baselines on WINE dataset and E-MSA on CIFAR-10. We also note that DDPNOpt admits relatively small variance compared with OCP-inspired baselines.
>
> ****
>
> [1] https://datascience.stackexchange.com/questions/20179/what-is-the-advantage-of-keeping-batch-size-a-power-of-2

---

> > ### Author Response · Authors · 2020-11-18
> > **Author Response to Reviewer 2 (Part 2/3)**
> >
> > **6. $V_{xx}$ regularization in Fig 4**
> > - This corresponds to the Tikhonov regularization that is added to the value Hessian during DDPNOpt backward pass, _i.e._ $V_{xx} \leftarrow V_{xx} + \epsilon I_t$, where $I_t$ is the identity matrix and $\epsilon$ is the value reported in the axis of Fig 4. This regularization appears in previous DDP literature for stabilizing complex humanoid robotics [2]. For the application of DNN training, we find the regularization useful in preventing the value Hessian from low rank when the dimension of the hidden state varies along propagation, which tends to improve the robustness. In the revision, we include an additional paragraph discussing its effect (page 6) with the pseudo code of DDPNOpt (Alg 3).
> >
> > ****
> >
> > **7. Clarification on Fig 4**
> > - In Tables 9 and 10 (Appendix A.7.4), we report the _'absolute'_ numerical values for this experiment. Computing the performance difference between DDPNOpt and baselines from Tables 9 and 10 reproduces Fig 4a. Since different learning rates greatly affect the resulting training loss/accuracy, visualizing these values on the same plot as in Fig 4a conveys little information. We note that while the accuracy for baselines tends to drop dramatically (e.g. 91 -> 55 for SGD) as the learning rate increases, the accuracy for DDPNOpt stays relatively close (93 -> 70).
> > - In Appendix A.7.3, we report the numerical values when each baseline uses its best-tuned learning rate (which is the values we report in Table 3) and compare with its DDPNOpt counterpart using the same learning rate. As shown in Tables 6,7,8, for most cases extending the baseline to accept the Bellman framework improves the performance. All values in Appendix A.7.3 and A.7.4 are averaged over 10 seeds.
> > - Fig 4a and 4b aim to demonstrate different aspects of the performance between DDPNOpt and baselines. In Fig 4a, we report the performance difference of the _final_ training results. On the other hand, Fig 4b discusses the convergence dynamics over the entire optimization process with two case studies. The upper part of Fig 4b demonstrates DDPNOpt has stabler convergence in both loss/accuracy, while the bottom part showcases an example where the final training result looks similar (also indicated in Fig 4a) but DDPNOpt converges faster compared with the baseline. In practice, the difference in convergence behavior is more obvious for SGD compared with RMSprop/EKFAC.
> > - Lastly, we correct parts of Fig 4a in the revision as we identified misleading values for two sets of learning rate (RMSprop lr=0.045, $V_{xx}$=1e-5 and EKFAC lr=0.3). We stress that all of our claims remain valid and all figures are generated from the same submission package.
> >
> > ****
> >
> > **8. Procedure to generate Fig 5**
> > - First, we perform standard DDPNOpt steps to compute layer-wise policies $(k_t, K_t)$. Next, we conduct singular-value decomposition on the feedback matrix $K_t$. In this way, the leading right-singular vector corresponding to the dominating $\delta x_t$ that the feedback policy shall respond with. Since this vector is with the same dimension as the hidden state $\mathbb{R}^{n_t}$, which is most likely not the same as the image space, we project the vector back to image space using the techniques proposed in [3]. The pseudo code and computation diagram are included in Appendix A.7.1.
> >
> > ****
> >
> > **9. Vanishing gradient**
> > - In Fig 11, we provide additional experiments on vanishing gradient. Fig 11a reports the training result with the same Sigmoid network but using the CE loss (notice the numerical differences in the y-axis for different objectives). None of the presented optimizers were able to escape from VG, as evidenced by the degenerate update magnitude.
> > - On the other hand, changing the activation to ReLU eliminates VG (Fig 11b). While SGD-VGR, EKFAC, and DDPNOpt can all escape from VG with ReLU network, DDPNOpt admits slightly faster convergence with smaller variance compared with the baselines. We are not able to obtain the results for DDPNOpt2nd in this experiment setup. We refer the reviewer to  _**11. Behavior of DDPNOpt2nd with ReLU network**_ for the discussion.
> > - Therefore, in Fig 6 we report the results with MMC loss since the loss encourages the feature points to spread more widely over the feature space, leading to a denser Hessian spectrum. Although this is originally proposed to improve the robustness against adversarial attacks, it is also suitable in our application since the feedback policies are directly computed from $V_{xx}$. For a pictorial illustration of the difference between MMC and CE loss on a low-dimensional space, we refer the reviewer to Fig 1 in [4].
> >
> > ****
> >
> > [2] Synthesis and stabilization of complex behaviors through online trajectory optimization
> > [3] Visualizing and understanding convolutional networks.
> > [4] Rethinking Softmax Cross-Entropy Loss for Adversarial Robustness

---

> > > ### Author Response · Authors · 2020-11-18
> > > **Author Response to Reviewer 2 (Part 3/3)**
> > >
> > > _**Other clarification**_
> > >
> > > **10. Matrix-free solver on $Q^t_{uu}$**
> > > - The dimension of $Q^t_{uu}$ is indeed quadratic in the number of parameter of a single layer. In the revision, we equip vanilla DDP with the matrix-free solver and report new results in Table 3. For the datasets trained with feedforward networks, we are able to have vanilla DDP run on all related datasets. While vanilla DDP performs better compared with E-MSA on (F)MNIST, our DDPNOpt and standard baselines still lead these datasets by a margin. In practice, we notice that running vanilla DDP on these larger datasets often admits unstable training; hence requires careful tuning on the hyper-parameters. As for CNN-related datasets, which involve a much larger batch size (128) and hidden state (usually 100x larger than feedforward networks), we encounter memory explosion when computing the value Hessians and the feedback policies (max memory is 24G on our NV RTX TITAN). However, improvement may be possible as our implementation may not fully optimize the intermediate memory usage. This is of independent interest for future investigations.
> > >
> > > ****
> > >
> > > **11. Behavior of DDPNOpt2nd with ReLU network**
> > > - We first restate the layer propagation rule: $x_{t+1} = \sigma_t (g_t(x_t,u_t))$. When the activation function $\sigma_t$ has degenerate second-order derivatives, the only second-order term left after the expansion of the dynamics $f_t \equiv \sigma_t \circ g_t$ will be $g^t_{ux}$. We note that since the affine transmission $g_t$ is bilinear in $x_t$ and $u_t$, $g^t_{xx}$ and $g^t_{uu}$ vanish regardless of the expanding order. The presence of $g^t_{ux}$ can sometime make the training unstable, especially when $\sigma_t$ is unbounded activation function like ReLU. This is because the tensor $g^t_{ux}$ contains only the value '1's placing at certain locations (please refer to Eq 28 in Appendix A.5). Hence, when computing $Q^t_{ux}$, the tensor product $V^t_h \cdot g^t_{ux}$ can easily blow up if $V^t_h$ is too large (which is likily for ReLU since it's unbounded).
> > >
> > > ****
> > >
> > > **12. Over-parametrized $f_t$**
> > > - As pointed out by the reviewer, _'highly over-parametrized'_ implies the parameter of each layer is in high dimension space. The description is typically used in deep learning literature [5-6] to indicate an optimization scheme where the dimension of the parameter is much higher than the data size. We include both descriptions in the revision to avoid confusion.
> > >
> > > ****
> > >
> > > **13. Clarification on Fig 1**
> > > - In the original figure, the red arrow in weight update phase indicates the computation $u \leftarrow u +$'update', where the 'update' comes from from either Backprop $\delta u_t^*$ or DDP $\delta u_t^*(\delta x_t) = k_t + K_t(\delta x_t)$. The faded arrows are meant to indicate that Backprop applies the weight update sequentially from the initial layer (line 7-9 in Alg 2) except without any dependency from previous layers. This is in contract to DDPNOpt which requires computing $\delta x_t$ from previous layers; hence we use the solid arrows instead. Due to space constraint, we only specify the key equations that are involved during backward pass and instead highlight the differences in the quantities being carried to the preceding layer ($\nabla_{x_t}J$ vs $(V^t_x,V^t_{xx})$).
> > > - Since the purpose of Fig 1 is to provide pictorial aids when comparing Alg 1 and 2, in the revision we simplify the expression and provide additional descriptions in the legend. Additionally, Fig 1 is now placed beside Proposition 2 where we discuss the connection between DDP and Back-propagation, so that the related notations are properly introduced.
> > >
> > > ****
> > >
> > > **14. Notation**
> > > - As pointed out by the reviewer, we use $\mathcal{F}_t$ to denote any real-valued time-dependent function.
> > > - $\nabla_{x_t} \mathcal{F}_t \equiv \mathcal{F}^t_x$  denotes the first-order derivative of $\mathcal{F}_t$ evaluated on a given point $x_t$. The abbreviation $\mathcal{F}^t_x$ is made so as to align with the notational convention in the DDP literature [7-9]. We have clarified these notations in the revision. The definition of the vector-tensor product (footnote 1 on page 3) and explicit specification of the evaluating point (page 2 and 3) are also included.
> > >
> > > ****
> > >
> > > [5] Empirical Analysis of the Hessian of Over-Parametrized Neural Networks
> > > [6] How SGD Selects the Global Minima in Over-parameterized Learning: A Dynamical Stability Perspective
> > > [7] Synthesis and Stabilization of Complex Behaviors through Online Trajectory Optimization
> > > [8] Receding Horizon Differential Dynamic Programming
> > > [9] Control-Limited Differential Dynamic Programming

---

> > > > ### Comment · AnonReviewer2 · 2020-11-21
> > > > **Thank you for the clarifications!**
> > > >
> > > > I appreciate the detailed response. All of my major concerns have been addressed. The authors have made some good changes to the paper and, at this point, I don't see any notable issues remaining. I will be updating my score to reflect that.

---

### Author Response · Authors · 2020-11-18
**Author response to all reviewers**

We thank the reviewers for their valuable comments. We are excited that the reviewers identified the novelty of the connection (R1,R2,R3,R4), comprehensive experiments (R3,R4), and well-written presentation (R1,R2,R3,R4) of our work. We believe DDPNOpt takes a significant step toward new algorithmic design inspired from optimal control. We address raised concerns and **revise the submission with the modifications marked as blue**. Additional experiments and discussions raised by each reviewer are left in **Appendix A.7**. Below we summarize the changes to the paper organization in the revision.

1. Following R4's suggestion on increasing the reading experience for a broader audience, we introduce DDPNOpt right after we present the connection between vanilla DDP and existing methods (Proposition 2), so that the derivation of DDPNOpt is close to DDP, making it easier to compare between three frameworks. Discussion on _'Role of the feedback $K_t$ and $\delta x_t$'_ (_i.e._ the latter part of Sec 3 in the original submission) now appears after the derivation of DDPNOpt.

2. The pseudo code of DDPNOpt is now attached on page 6 (Alg. 3), with the computation at each stage linked to the corresponding equation.

3. Fig 1 is now placed beside Proposition 2 where we discuss the connection between DDP and Back-propagation, so that the notations in the figure can be properly introduced. We also simplify the expression and provide additional descriptions in the legend.

---

### Decision · Program_Chairs · 2021-01-07
**Final Decision**

**Decision:**

Accept (Spotlight)

**Comment:**

Reviewers agreed that connecting neural networks with dynamical systems to create a new kind of optimizer is an interesting idea. After the authors' improvements, this is a strong submission of wide interest.